# Seasonal dynamics and molecular phylogenetic studies on cercariae in Central Zone of Kashmir valley

Zahoor Ahmad Wani[1]*, Rafiq A. Shahardar[1], Kamal H. Bulbul[1], Idrees M. Allaie[1], Showkat A. Shah[2], Shabir A. Rather[3], Aiman Ashraf[1], Shahana Tramboo[1] Asif H. Khan[1]

1 Division of Veterinary Parasitology, FVSc and AH, SKUAST-Kashmir, Srinagar, Jammu and Kashmir, India,
2 Division of Veterinary Pathology, FVSc and AH, SKUAST-Kashmir, Srinagar, Jammu and Kashmir, India,
3 Department of Zoology, Baba Ghulam Shah Badshah University, Rajouri, Jammu and Kashmir, India

* zahoorwani_103@yahoo.co.in

## Abstract

A total of 12103 snails examined in Central Kashmir for determining the population dynamics of cercariae revealed overall prevalence to be 4.03%. Gymnocephalus (0.13%), furcocercous (0.28%), echinostome (0.34%) and xiphido-cercaria (3.26%) were recorded. The prevalence of cercaria was recorded highest in summer (4.28%) followed by spring (4.05%) and autumn (3.32%). None of the cercaria was recorded during winter season. Morphologically identified cercariae of Veterinary importance were subjected to molecular analysis using genus (28S rDNA and ITS-2) specific primers. The isolates of gymnocephalus cercaria (FA28, FC28, FZ28, FC2) were identified as cercarial stages of *Fasciola* spp. The phylogenetic trees revealed that the isolates FA28 and FC28 belonged to *Fasciola gigantica* and FZ28 and FC2 isolate to *F. hepatica*. The isolate of *Fasciola gigantica* (FA28, FC28, FZ28) showed 2, 4 and 13 nucleotide polymorphisms. There was addition and deletion of 1 and 8 nucleotides at various positions in case of *Fasciola* isolates respectively. Besides this, there were nucleotide substitutions at 4 positions along with presence of nucleotide T at 475 position which confirmed it be *Fasciola hepatica*. The isolates of echinostome cercaria (B1, BD13 and GY) were identified as cercarial stages of *Moliniella anceps*, *Echinoparyphium recurvatum* and family Echinostomatidae respectively. The *Moliniella anceps* isolate showed prominent differences at 8 positions with respect to other Echinostomatidae spp. The insertion of C at position 612 confirmed it to be *Moliniella anceps*, while as other two isolates showed 2 nucleotide polymorphisms each after 28S gene amplification. On ITS-2 rDNA analysis, the isolate B1 showed 7 nucleotide polymorphisms and phylogenetic tree revealed that the isolate B1, also belonged to *Echinoparyphium recurvatum*. The study made it very clear that molecular characterization employing internal transcribed spacer (ITS-2) and 28S ribosomal DNA sequences are reliable approach for genetic differentiation of cercarial stages

**Data availability statement:** All relevant data are within the manuscript.

**Funding:** The author(s) received no specific funding for this work.

**Competing interests:** The authors have declared that no competing interests exist.

of trematodes. The phylogenetic taxonomy of echinostomes is still unclear and molecular diversity found in this study is perhaps the first study from India as well as in Indian subcontinent. So, focus should be made more on echinostomes for understanding their morphological, biological and molecular diversity for clarifying their taxonomic position.

## Introduction

Parasitic diseases impose a severe economic burden on livestock in the form of morbidity, mortality, reduced feed efficiency and by way of costs incurred on treatment and control [1]. Among trematodes, the world-wide losses in animal productivity due to fasciolosis were estimated to be $ 3.2 billion per annum through the death of animals, liver condemnation and productivity losses and is considered as one of the significant helminth parasites [2]. Deaths due to immature paramphistomes may be as high as 80–90% in domesticated ruminants [3] and schistosomosis is a threat to 530 million cattle infecting over 165 million cattle in Africa and the Middle East [4].

The Digenian trematodes have complicated lifecycle in which molluscs play a key role as intermediate hosts. The typical digenean life cycle involves two free living, the miracidium and cercaria and four parasitic stages, the sporocyst and the redia, metacercaria and adult. Temperature and vegetation play definite role in the growth and maturation of snails as a result their population would follow a definite seasonal trend. Increase and decrease in snail population will directly affect the cercarial output from them. In addition to it, snails undergo hibernation when there is increase in temperature above $30^0C$ and decrease in temperature below $10^0C$. Temperature increases the emergence of cercariae from the snail and the acceleration of their production within them [5]. Trematodes in fresh water environment have been studied for nearly three (03) centuries. Classification of digeneans is a complicated task, but the larval characters of the digeneans can be used in classification. Sewell [6] studied the fresh water cercariae from India and he modified the Luhes [7] classification and divided major groups into a number of smaller groups. After Sewell, few workers have described some cercariae from India [8,9].

The conventional methods used to examine cercarial infection in snails are typically performed by exposing the snails to light (shedding) and/or dissection (crushing). This method considers only the morphological characteristics which consumes more time and requires a high level of experience-based skill. The identification of cercariae becomes difficult due to their morphological similarity with other related species along with presence of soft stable features that are subjected to host induced phenotypic variations [10,11]. Nowadays, molecular biological approaches like PCR have been applied for the accurate identification of cercariae which has made it easy to investigate the epidemiology of fluke infections [12,13] and has enabled the identification of species and screening of genetic variants among population [14–16]. Several molecular markers like ITS-1, ITS-2, 18S and 28S of nuclear rDNA and CO-1 of mitochondrial DNA have been shown to be reliable genetic markers for species

identification and for determining intra/inter-specific polymorphisms in the parasites [17,18]. There is scanty literature available regarding molecular characterization of cercariae of different digenetic trematodes in India [19–21] and no such study has been carried out in Kashmir Valley. Among trematodes, *Fasciola gigantica*, *Fasciola hepatica*, *Dicrocoelium dendriticum*, *Paramphistomum cervi*, *Cotylophoron cotylophorum*, *Gastrothylax crumenifer* and *Carmyerius spatiosus* have been reported by different workers from various parts of Jammu and Kashmir based on faecal and necropsy studies [22–24], but no study has been carried out to identify the cercarial stages of above-mentioned trematodes. Apart from the morphological features, there is also a need to confirm the cercarial fauna by molecular techniques to understand their phylogenetic relationship with their counterparts in other regions of the world. Therefore, the study was undertaken to study the prevalence of various types of cercariae present in the snails and validate morphologically identified cercariae of Veterinary importance by DNA sequencing of 28S and ITS-2 region of nuclear ribosomal DNA.

## Materials and methods

### Location and geography of study area

The study was conducted in Central Zone of Kashmir Valley which comprises of 3 districts *viz*. Budgam, Ganderbal and Srinagar. The geographical locations of study sites are provided in Table 1 using GPS Map Camera (version 1.6.33). In Central Kashmir, summers are usually mild with good little rain, but relative humidity is generally high and nights are cool. The precipitation occurs throughout the year and no month is particularly dry. The hottest month is July (maximum temperature of 32°C and minimum temperature 6°C) and the coldest are December-January (maximum temperature 0°C and minimum temperature −1.5°C).

### Prevalence of cercariae

**Collection and identification of snails.** Snails were collected from ponds, lakes, marshy areas, temporary water bodies, paddy fields, stagnant water bodies, of each of these districts in each season of the year. Identification of representative specimens was got confirmed from Zoological Survey of India, Kolkata (**ZSI, Moll: I.R.No.107**) and Department of Parasitology, College of Veterinary Sciences, Assam Agricultural University, Khanapara.

**Table 1. Detailed information about geographical locations of study areas.**

| S.No. | Collection Site Name | District | Latitude | Longitude | Altitude (m) |
|---|---|---|---|---|---|
| 1 | Narkar | Budgam | 34.040611 | 74.753519 | 1761 |
| 2 | Chadoora, Hisipora | Budgam | 33.943094 | 74.811091 | 1640 |
| 3 | Narbal | Budgam | 34.117030 | 74.672822 | 1607 |
| 4 | Ompora | Budgam | 34.030922 | 74.738583 | 1619 |
| 5 | Ranger | Budgam | 33.939424 | 74.767321 | 1709 |
| 6 | Rangil | Ganderbal | 34.206752 | 74.806955 | 1647 |
| 7 | Shuhama | Ganderbal | 34.188608 | 74.828563 | 1651 |
| 8 | Pandach | Ganderbal | 34.216505 | 74.771891 | 1608 |
| 9 | Beehama | Ganderbal | 34.216163 | 74.781352 | 1609 |
| 10 | Rakhrabitar | Ganderbal | 34.205961 | 74.719883 | 1606 |
| 11 | Hyderpora | Srinagar | 34.048247 | 74.783444 | 1604 |
| 12 | Qamarwari | Srinagar | 34.102819 | 74.764358 | 1604 |
| 13 | Hazratbal | Srinagar | 34.129830 | 74.837552 | 1611 |
| 14 | Bemina | Srinagar | 34.078661 | 74.765891 | 1605 |
| 15 | Nishat | Srinagar | 34.132380 | 74.877738 | 1620 |

**Screening of snails for cercarial shedding.** The collected snails were screened initially by putting 10 individuals in one 200 ml disposable plastic glass, exposed them to sunlight to facilitate shedding of cercariae. Thereafter, the snails found positive for cercariae, were then placed individually in transparent glass container so as to observe the release of cercariae. Cercarial shedding was observed in all the four seasons of the year for evaluating the seasonal prevalence.

**Crushing method.** Snails which did not shed cercariae by shedding method, about 5% of the respective species were crushed to observe the presence of any larval stage in them. The pattern was followed for all the seasons of the year.

**Staining of cercariae by Lugol's iodine stain.** One drop of water containing 25–30 freshly released cercariae was placed on a glass slide. The slides were then heated over a spirit lamp to kill them. A drop of 2% Lugol's iodine was added gently and a coverslip was put on the stained water. The slides were then examined for identification under 100x and 400x magnifications. Photographs were also taken to record different types of cercariae observed during the study programme.

**Identification of cercariae.** The stained cercariae were identified as per the standard keys [25–27]. The remaining cercariae were then kept in 70% alcohol or deep refrigeration (-20°C) for further molecular studies.

## Statistical analysis

The results were subjected to standard statistical analysis [28] . The Z-test applied on prevalence data was referred for p-values for its significance. Any p-value less than 0.05 ($p \leq 0.05$) was taken as statistically significant.

## Molecular identification of morphologically identified cercaria

Morphologically identified cercariae were subjected to molecular identification using ITS-2 and 28S rDNA gene specific primers of each trematode parasite.

**Extraction of Genomic DNA.** Morphologically identified cercariae were grouped into gymnocephalus, echinostome and furcocercous cercariae for DNA extraction. Also, snails suspected for amphistome larval stage/s were taken and DNA was extracted as per manufacturer's protocol of DNeasy® Blood & Tissue Kit (QIAGEN). Approximately 25–30 mg homogenized cercariae of each group released by particular snail and 1 gram of snail tissue (in case of amphistome) was taken.

**Purity and concentration of genomic DNA.** The purity was estimated by determining the ratio of A260/A280, which for pure sample falls between 1.7 and 1.9 using a Nano-drop instrument.

**Polymerase chain reaction.** In the present study, one region of the ribosomal DNA located as internal transcribed spacer 2 (ITS-2) and part of 28S rDNA gene were amplified. The primer sequences, reaction mixture, cycling conditions and expected size of amplified product for 28S rDNA and ITS-2 region for each parasite possessing these cercarial types as life cycle stages are shown in Tables 2 and 3.

**Table 2. PCR reaction mixture for amplification of 28S and ITS-2 region of rDNA of Trematode parasite from respective cercariae.**

| S. No. | *Fasciola* from gymnocephalus cercaria | | | *Schistosoma* from furcocercous cercaria | | | Echinostomatidae family from echinostome cercaria | | | Paramphistomitidae family from amphistome larval stage | | |
|---|---|---|---|---|---|---|---|---|---|---|---|---|
| | Mixture component | 28S (µl) | ITS-2 (µl) | Mixture component | 28S (µl) | ITS-2 (µl) | Mixture component | 28S (µl) | ITS-2 (µl) | Mixture component | 28S (µl) | ITS-2 (µl) |
| 1 | PCR Buffer | 2.5 | 2 | PCR MM | 12 | 12 | PCR Buffer | 2.5 | 2.5 | PCR Buffer | 2.5 | 2 |
| 2 | MgCl$_2$ | 2 | 1.2 | Forward primer | 1 | 1 | MgCl$_2$ | 2 | 2 | MgCl$_2$ | 2 | 1.2 |
| 3 | DNTPs | 0.5 | 0.4 | Reverse primer | 1 | 1 | DNTPs | 0.5 | 0.5 | DNTPs | 0.5 | 0.4 |
| 4 | Forward primer | 0.5 | 1.25 | Taq polymerase | 0.5 | 0.5 | Forward primer | 0.75 | 0.75 | Forward primer | 1 | 1.0 |
| 5 | Reverse primer | 0.5 | 1.25 | DNA | 1 | 1 | Reverse primer | 0.75 | 0.75 | Reverse primer | 1 | 1 |
| 6 | Taq polymerase | 0.3 | 0.3 | Nuclease free water | 19.5 | 19.5 | Taq polymerase | 0.2 | 0.2 | Taq polymerase | 0.3 | 0.3 |
| 7 | DNA | 1 | 1 | | | | DNA | 1 | 1 | DNA | 1 | 1 |
| 8 | Nuclease free water | 17.7 | 12.6 | | | | Nuclease free water | 17.3 | 17.3 | Nuclease free water | 16.7 | 13.1 |
| | **Total** | **25** | **20** | | **25** | **25** | **Total** | **25** | **25** | **Total** | **25** | **20** |

**Table 3. Primer sequences, cycling conditions and the expected size of amplified products of various parasites of Veterinary importance having these cercarial types as life cycle stages.**

| Parasite (Cercaria) | Primer sequence (5'-3') | Region amplified/ Target gene | Cyclic conditions | | Product size (bp) | Reference |
|---|---|---|---|---|---|---|
| *Fasciola* spp. (Gymnocephalus cercaria) | F: ACG TGA TTA CCC GCT GAA CT<br>R: CTG AGA AAG TGC ACT GAC AAG | 28S rDNA | Initial denaturation<br>Cyclic denaturation<br>Annealing<br>Extension<br>Final Extension | 94°C- 3 min.<br>94°C- 30 sec. ⎫<br>60°C- 30 sec. ⎬ 30 cycles<br>72°C- 60 sec. ⎭<br>72°C- 5 min. | 618 | [29] |
| | F: GGTGGATCACTGGGCTCGTG<br>R: TATGCTTAAATTCAGCGGGT | ITS-2 | Initial denaturation<br>Cyclic denaturation<br>Annealing<br>Extension<br>Final Extension | 95°C- 5 min.<br>95°C- 30 sec. ⎫<br>60°C- 30 sec. ⎬ 37 cycles<br>72°C- 45 sec. ⎭<br>72°C- 5 min. | 550 | [30] |
| *Schistosoma* spp. (Furcocercous cercaria) | F: ACC CGC TGAATT TAA GCA<br>R: TCC TGA GGG AAA CTT CGG | 28S rDNA | Initial denaturation<br>Cyclic denaturation<br>Annealing<br>Extension<br>Final Extension | 95°C- 5 min.<br>94°C- 1 min. ⎫<br>50°C- 2 min. ⎬ 30 cycles<br>72°C- 3 min. ⎭<br>72°C- 10 min. | 1225 | [31] |
| | F: CGG TGG ATC ACT CGG CTC<br>R: -CCT GGT TAG TTT CTT TTC CTC CGC | ITS-2 | Same as that of 28S rDNA | | 501 | [32] |
| *Echinostoma* spp. (Echinostome cercaria) | F: GTA CCG TGA GGG AAA GTT G<br>R: GTC CGT GTT TCA AGA CGG G | 28S rDNA | Initial denaturation<br>Cyclic denaturation<br>Annealing<br>Extension<br>Final Extension | 94°C- 5 min.<br>94°C- 30 sec. ⎫<br>50°C- 30 sec. ⎬ 30 cycles<br>72°C- 45 sec. ⎭<br>72°C- 5 min. | 554 | [31] |
| | F: CGG TGGATCACTCGGCTCGT<br>R: CCTGGTTAGTTTCTTTTCCTCCGC | ITS-2 | Same as that of 28S rDNA | | 588 | [32] |

*(Continued)*

**Table 3.** (Continued)

| Parasite (Cercaria) | Primer sequence (5'-3') | Region amplified/ Target gene | Cyclic conditions | | Product size (bp) | Reference |
|---|---|---|---|---|---|---|
| **Paramphistomitidae** (Amphistome cercaria) | **F:** AAG CAT ATC ACT AAG CGG <br> **R:** GCT ATC CTG AGG GAA ACT TCG | **28S rDNA** | Initial denaturation <br> Cyclic denaturation <br> Annealing <br> Extension <br> Final Extension | 95°C- 5 min. <br> 94°C- 1 min. <br> 56°C- 1 min. <br> 72°C- 1 min. <br> 72°C- 10 min. (30 cycles) | 1200 | [33] |
| | **F:** AGA ACA TCG ACA TCT TGA AC <br> **R:** TAT GCT TAA ATT CAG CGG GT | **ITS-2** | Initial denaturation <br> Cyclic denaturation <br> Annealing <br> Extension <br> Final Extension | 94°C- 5 min. <br> 94°C- 30 sec. <br> 50°C- 30 sec. <br> 72°C- 45 sec. <br> 72°C- 5 min. (37 cycles) | 500 | [18] |

**PCR product analysis.** The amplified PCR products were checked by running 10 µl of PCR products plus 3 µl of loading dye in 1% agarose (w/v) in horizontal gel electrophoresis in 1X TAE buffer at 90 V for 45 minutes. 3 µl of a 100 bp DNA ladder was also run alongside the samples to ascertain the size of the amplified products. After electrophoresis, the amplified products were visualized under UV light in a UV trans-illuminator and documented in a gel documentation system.

**Sequencing of the 28S rDNA and ITS-2 amplicons.** The randomly selected amplified PCR products from each cercarial type (representing each gene) were sent for DNA sequencing (from both the directions by primer walking method) to Xcelris, Ahmedabad, Gujarat, along with respective forward and reverse primers. The purification of the dispatched PCR products was done by the respective company.

**Species identification, divergence and phylogenetic analysis.** The raw sequence results obtained were edited using GENE TOOL software and the sequences were analyzed using BLAST [34] search of NCBI for determining the similarity with the sequences present in the nucleotide database. The sequences were further aligned using Multi align editor of GENE TOOL and MegAlign programme of DNA STAR softwares with published sequences of 28S rDNA and ITS-2 genes of respective trematode parasites. To know the accuracy of divergence, the phylogenetic trees were constructed by two approaches *viz.* maximum likelihood approach and neighbour joining approach separately as well as jointly for all the cercarial isolates (representing both genes) using Mega 6 software.

## Results

### Morphological identification of cercariae

The different cercariae grouped into gymnocephalus, echinostome, furcocercous and xiphido cercariae were identified on the basis of following morphological features.
**Gymnocephalus cercaria:** They possess both oral and ventral sucker, but ventral sucker is situated slightly behind the middle of the body. Such cercariae have a simple and long tail (Fig 1).
**Echinostome cercaria:** They possess 2 suckers, out of which oral sucker is characterized by the presence of single or double row collar of spines around it (Fig 1).
**Furcocercous cercaria:** They are characterized by a forked tail and some species may even possess eyespots. They are of two types, i.e., brevicaudum longifurcous (Avian spp.) and longicaudum brevifurcous (mammalian spp.) (Fig 1).
**Xiphido-cercaria:** The cercariae possess body which is oval shaped, both oral and ventral sucker are present, the oral sucker being equipped with a stylet (Fig 1).

### Overall prevalence of cercariae in snails

Overall prevalence of cercariae was found to be 4.03% and 4 types of cercariae recorded were gymnocephalus cercaria, echinostome cercaria, furcocercous cercaria and xiphido-cercaria (Fig 1). Highest prevalence was recorded for xiphido-cercaria (3.26%) followed by echinostome cercaria (0.34%), furcocercous cercaria (0.28%) and lowest for gymnocephalus cercaria (0.13%). The difference was found statistically significant (p < 0.05) between all the cercariae except for echinostome and furcocercous cercariae (Table 4).

Non-significantly (p > 0.05) highest prevalence of cercariae was observed in District Budgam (4.11%), followed by Ganderbal (4.23%) and Srinagar (3.57%). In all the 3 districts, highest prevalence was observed for xiphido-cercaria followed by echinostome cercaria, furcocercous cercaria and lowest for gymnocephalus cercaria. None of the snails released amphistome cercaria (Table 4).

### Prevalence of cercariae in different snails

The prevalence of echinostome cercaria in *Lymnaea stagnalis*, *L. lagotis*, *L. brevicauda* and *Gyraulus ladacensis* was observed as 0.56%, 1.22%, 0.73% and 0.07% respectively. Furcocercous cercaria was recorded to be 1.61%, 0.32%

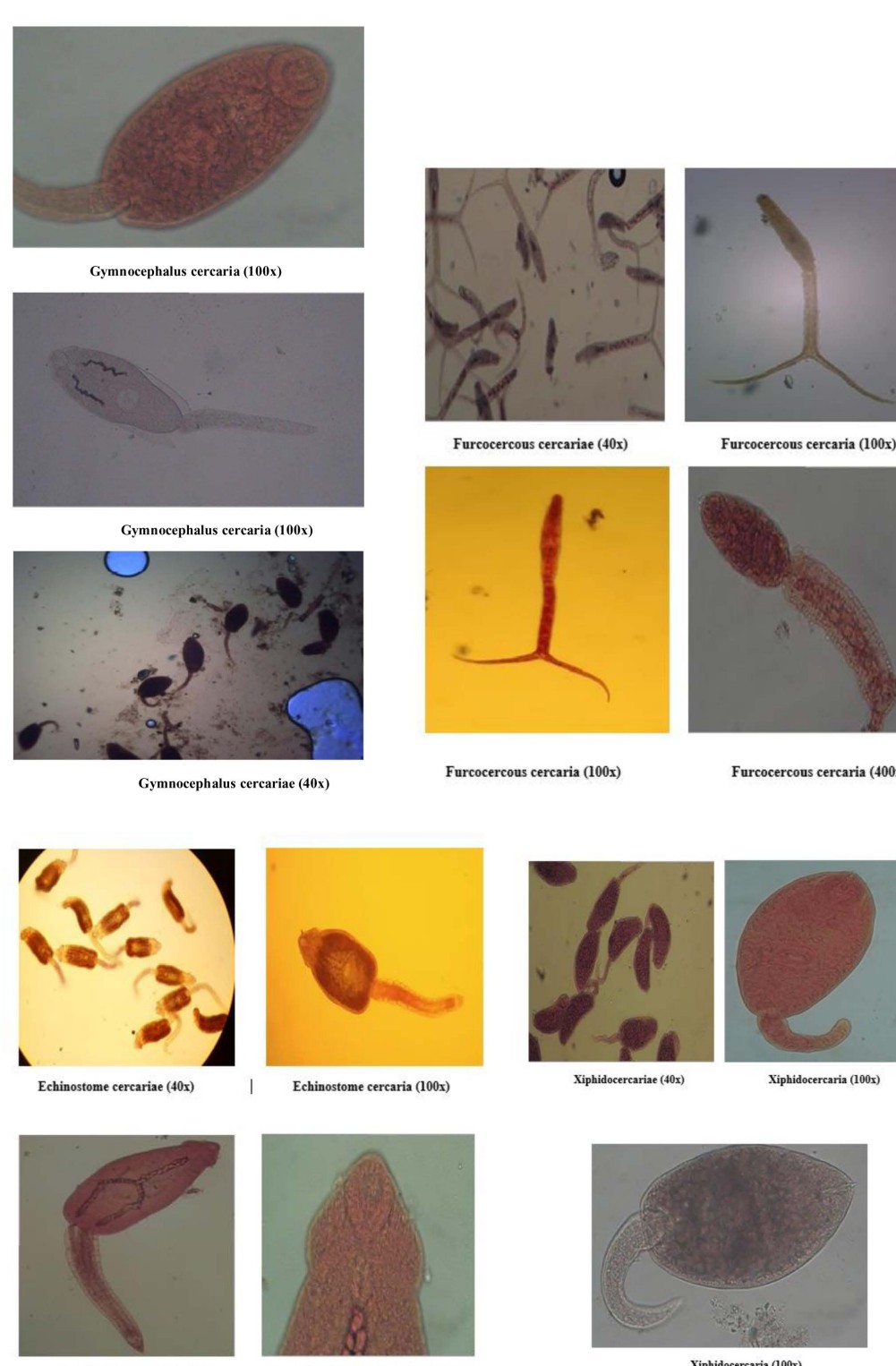

**Fig 1. Photographs showing different types of cercariae based upon the morphological features.**

**Table 4. Overall prevalence of cercariae in snails.**

| Area screened | Snails Collected | Snails Positive | Gymnocephalus cercaria | Echinostome cercaria | Furcocercous cercariae | Xiphido-cercaria |
|---|---|---|---|---|---|---|
| Budgam | **3961** | 163 (4.11)[aA] | 05 (0.12)[a] | 13 (0.32)[b] | 10 (0.25)[ab] | 135 (3.40)[c] |
| Ganderbal | **5152** | 218 (4.23)[aB] | 08 (0.15)[a] | 19 (0.37)[b] | 17 (0.33)[ab] | 174 (3.38)[c] |
| Srinagar | **2990** | 107 (3.57)[aC] | 03 (0.10)[a] | 10 (0.33)[b] | 08 (0.27)[ab] | 86 (2.88)[c] |
| **Grand Total** | **12103** | 488 (4.03) | 16 (0.13)[a] | 42 (0.34)[b] | 35 (0.28)[b] | 395 (3.26)[c] |

Figures in the parenthesis indicate % prevalence

Prevalence values (in parenthesis) of cercaria along a row bearing different small case superscript differ significantly

Prevalence values (in parenthesis) of cercaria along a column bearing different upper-case superscript differ significantly

and 0.21% prevalent in *L. stagnalis*, *L. lagotis* and *G. ladacensis* respectively. Gymnocephalus cercaria was released by *L. auricularia* snails only to the tune of 1.90%. Xiphido-cercaria was recorded as 12.37%, 3.68%, 3.66%, 3.08%, 6.18%, 0.41% and 0.44% prevalent in *L. stagnalis*, *L. luteola*, *L. lagotis*, *L. brevicauda*, *L. auricularia*, *B. troscheli* and *G. ladacensis* respectively. *I. exustus*, *G. pankogensis* and unidentified snails did not yield any cercaria (Table 5).

**Table 5. Prevalence of cercariae in different snails.**

| S. No. | Snail spp. | Snails collected | Snails Positive | Gymno-cephalus cercaria | Echinos-tome cercaria | Furcocercous cercaria | Xiphido-cercaria |
|---|---|---|---|---|---|---|---|
| 1 | *Lymnaea stagnalis* | **1611** | 234 (14.54)[E] | 0 (0.00)[A] | 9 (0.56)[B] | 26 (1.61)[B] | 199 (12.37)[D] |
| 2 | *Lymnaea Luteola* | **624** | 23 (3.68)[D] | 0 (0.00)[A] | 0 (0.00)[A] | 0 (0.00)[A] | 23 (3.68)[B] |
| 3 | *Lymnaea Lagotis* | **1882** | 98 (5.20)[D] | 0 (0.00)[A] | 23 (1.22)[B] | 6 (0.32)[B] | 69 (3.66)[B] |
| 4 | *Lymnaea brevicauda* | **1234** | 47 (3.81)[C] | 0 (0.00)[A] | 9 (0.73)[B] | 0 (0.00)[A] | 38 (3.08)[B] |
| 5 | *Lymnaea auricularia* | **841** | 66 (7.84)[D] | 16 (1.90)[A] | 0 (0.00)[A] | 0 (0.00)[A] | 52 (6.18)[c] |
| 6 | *Indoplanorbisexustus* | **644** | 0 (0.00)[A] | 0 (0.00)[A] | 0 (0.00)[A] | 0 (0.00)[A] | 0 (0.00)[A] |
| 7 | *Bithynia troscheli* | **969** | 4 (0.41)[B] | 0 (0.00)[A] | 0 (0.00)[A] | 0 (0.00)[A] | 4 (0.41)[A] |
| 8 | *Physa acuta* | **2258** | 10 (0.44)[B] | 0 (0.00)[A] | 0 (0.00)[A] | 0 (0.00)[A] | 10 (0.44)[A] |
| 9 | *Gyraulus ladacensis* | **1391** | 4 (0.29)[B] | 0 (0.00)[A] | 1 (0.07)[A] | 3 (0.21)[B] | 0 (0.00)[A] |
| 10 | *Gyraulus pankogensis* | **414** | 0 (0.00)[A] | 0 (0.00)[A] | 0 (0.00)[A] | 0 (0.00)[A] | 0 (0.00)[A] |
| 11 | **Unidentified Snails** | **235** | 0 (0.00)[A] | 0 (0.00)[A] | 0 (0.00)[A] | 0 (0.00)[A] | 0 (0.00)[A] |
| **Total** | | **12103** | 488 (4.03) | 16 (0.13)[a] | 42 (0.34)[b] | 35 (0.28)[b] | 395 (3.26)[c] |

## Overall seasonal prevalence of cercariae in snails

The highest prevalence was recorded in summer (4.28%) followed by spring (4.05%) and autumn season (3.32%), the difference being statistically significant (p < 0.05) between autumn and summer; spring and summer seasons ([Fig 2](link)). Gymnocephalus cercaria showed higher prevalence in autumn (0.21%) and lowest in summer (0.10%), while as echinostome cercaria was recorded in spring (0.42%) and lowest in autumn season (0.26%). Furcocercous cercaria was recorded highest in spring (0.33%) and lowest in autumn (0.26%), while as xiphido-cercaria showed higher prevalence in summer (3.56%) and lower prevalence in autumn season (2.58%). None of the cercariae was recovered in winter season ([Table 6](link)).

## District-wise seasonal prevalence of cercariae in snails

In District Budgam, the highest prevalence was recorded in summer followed by spring and autumn season, the difference being statistically non-significant (p > 0.05) between seasons In District Ganderbal, the highest prevalence was recorded in summer followed by spring and autumn season, the difference being statistically significant (p < 0.05) between autumn and spring; autumn and summer season and non-significant (p > 0.05) between spring and summer season, while in District

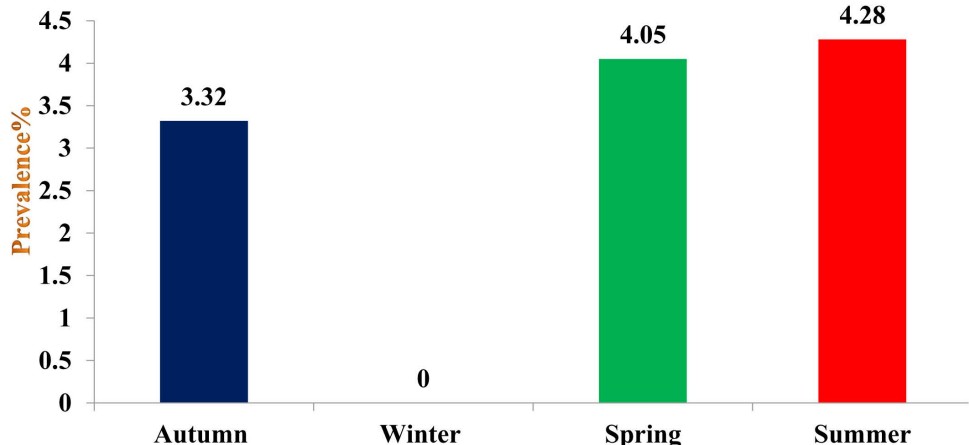

**Fig 2. Overall prevalence of cercariae in snails within different seasons of Central Zone of Kashmir Valley.**

**Table 6. Overall seasonal prevalence of cercariae in snails.**

| Season | Snails collected | Snails Positive | Gymnocephalus cercaria | Echinostome cercaria | Furcocercous cercaria | Xiphido cercaria |
|---|---|---|---|---|---|---|
| Autumn | 1896 | 63 (3.32)[AB] | 4 (0.21)[A] | 5 (0.26)[A] | 5 (0.26)[A] | 49 (2.58)[B] |
| Winter | 91 | 0 (0.00)[A] | 0 (0.00) | 0 (0.00) | 0 (0.00) | 0 (0.00) |
| Spring | 3603 | 146 (4.05)[B] | 5 (0.14)[A] | 15 (0.42)[B] | 12 (0.33)[A] | 114 (3.16)[C] |
| Summer | 6513 | 279 (4.28)[C] | 7 (0.10)[A] | 22 (0.33)[B] | 18 (0.27)[A] | 232 (3.56)[C] |
| Total | 12103 | 488 (4.03) | 16 (0.13)[a] | 42 (0.34)[b] | 35 (0.28)[b] | 395 (3.26)[c] |

Figures in the parenthesis indicate %prevalence

Prevalence values (in parenthesis) of cercaria across column bearing different upper-case superscript differ significantly

Srinagar, the highest prevalence was recorded in summer followed by spring and autumn season, the difference being statistically significant ($p < 0.05$) between autumn and spring; autumn and summer seasons and non-significant ($p > 0.05$) between spring and summer season (Figs 3–6, Table 7).

## Molecular identification of morphologically identified cercaria

**Gymnocephalus cercaria.** In order to confirm whether the morphologically identified gymnocephalus cercariae belonged to *Fasciola* spp. or not, they were subjected to molecular identification using gene (28S rDNA and ITS-2) specific primers as per Marcilla *et al*. (2002) and Raina *et al*. (2015), respectively.

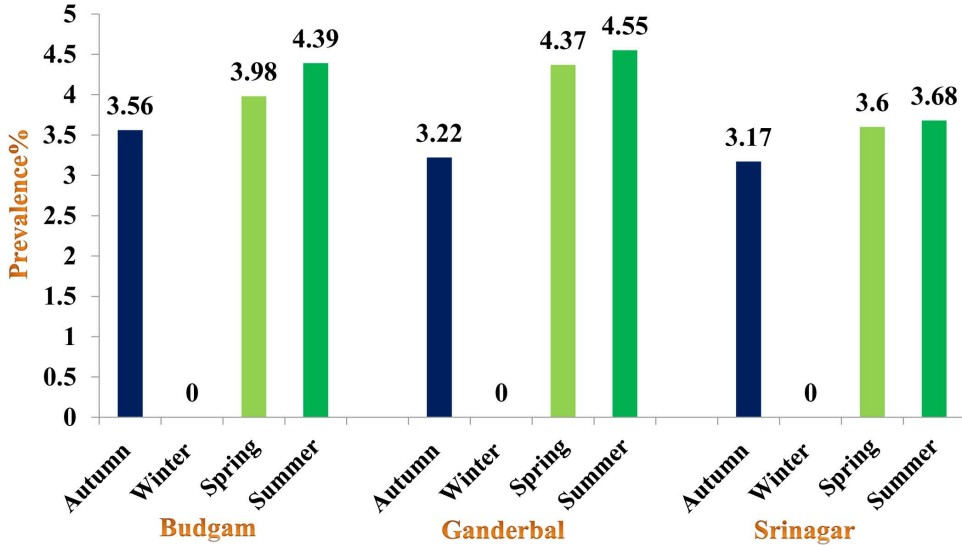

**Fig 3. Overall seasonal prevalence of cercariae in snails of different districts of Central Zone of Kashmir.**

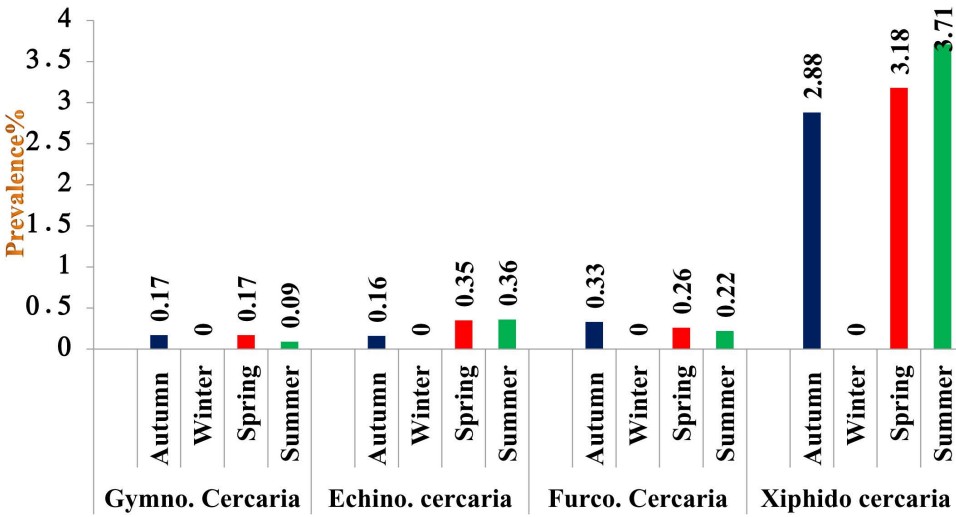

**Fig 4. Seasonal prevalence of different cercariae in district Budgam of Central Zone of Kashmir Valley.**

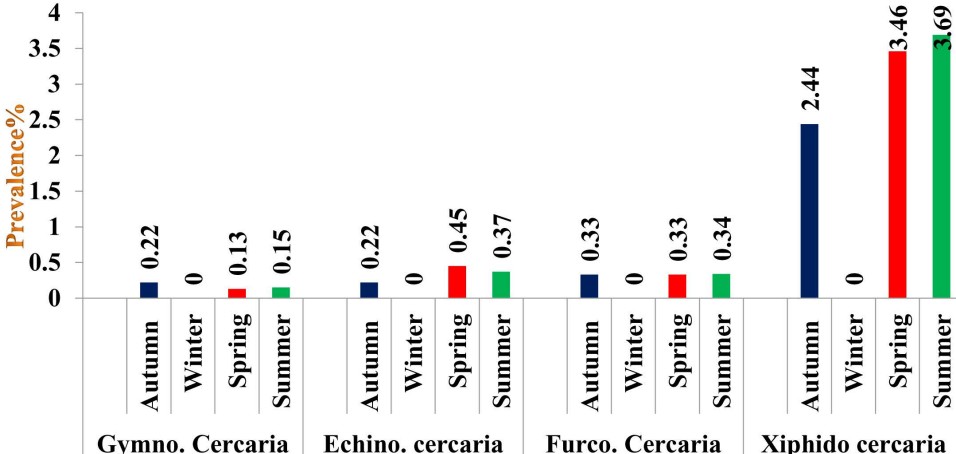

**Fig 5. Seasonal prevalence of different cercariae in district Ganderbal of Central Zone of Kashmir Valley.**

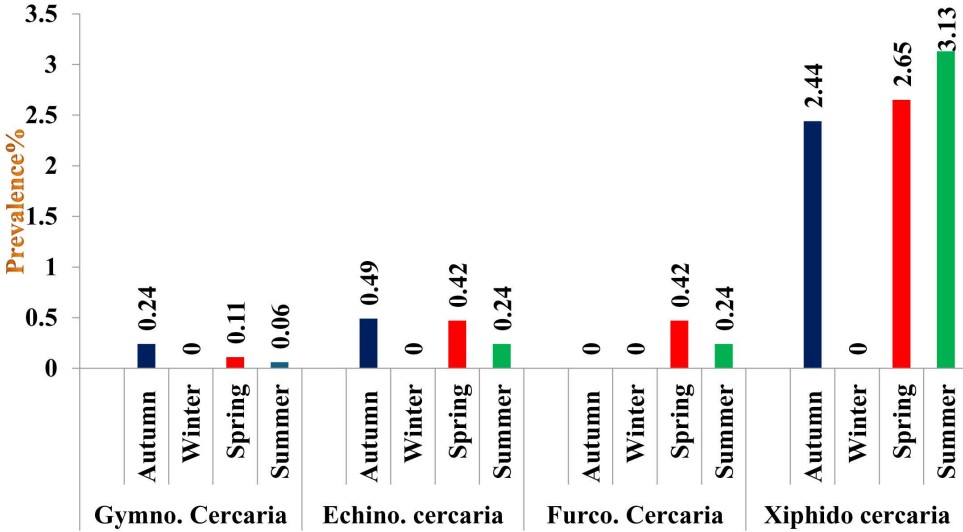

**Fig 6. Seasonal prevalence of different cercariae in district Srinagar of Central Zone of Kashmir Valley.**

**Amplification and nucleotide sequence analysis of 28S rDNA gene of *Fasciola spp.*** Out of 16 morphologically identified cercarial samples, 3 representative samples (FA28, FC28 and FZ28) were subjected to PCR which got amplified with expected product size of 618 base pair (Fig 7). The sequences of these amplicons were then aligned with the published sequences of 28S rDNA gene of *Fasciola* spp. (Accession Nos. AJ439739, AJ440785, HM126479, HM776945, JF323865, JF323866, KF791537, MF099787, MN148544 and AJ439738) (Fig 8).

The total length of the sequenced 28S rDNA amplicons of *Fasciola* isolates FA28 and FC28 were 619 bp, while for isolate FZ28, it was only 610 bp. The nucleotide sequence of FA28 isolate showed similarity ranging from 99.8% to 97.9% with the *Fasciola gigantica* Kashmir isolate (JF323866), *Fasciola gigantica* Bangalore (HM776945), Kolkata (HM126479), Izzatnagar (JF323865), Egypt (KF791537) and Vietnam (MF099787) isolates, *Fasciola gigantica* Santiago (AJ439739) and Burkina Faso (AJ440785) isolates, *Fasciola hepatica* Spain isolate (AJ439738) and *Fasciola gigantica* Karnataka isolate (MN148544)

**Table 7. District wise seasonal prevalence of cercariae in snails.**

| Area screened | Season | Snails Collected | Snails +ve | Gymnocephalus cercaria | Echinostome cercaria | Furcocercous cercaria | Xiphidocercaria |
|---|---|---|---|---|---|---|---|
| **Budgam** | **Autumn** | 589 | 21 (3.56)a | 1 (0.17)[a] | 1 (0.16)[a] | 2 (0.33)[a] | 17 (2.88)[b] |
| | **Winter** | 35 | 0 (0.00) | 0 (0.00) | 0 (0.00) | 0 (0.00) | 0 (0.00) |
| | **Spring** | 1129 | 45 (3.98)[a] | 2 (0.17)[a] | 4 (0.35)[a] | 3 (0.26)[a] | 36 (3.18)[b] |
| | **Summer** | 2208 | 97 (4.39)[a] | 2 (0.09)[a] | 8 (0.36)[a] | 5 (0.22)[a] | 82 (3.71)[b] |
| Total | | **3961** | 163 (4.11) | 05 (0.12)[a] | 13 (0.32)[b] | 10 (0.25)[ab] | 135 (3.40)[c] |
| **Ganderbal** | **Autumn** | 898 | 29 (3.22)[a] | 2 (0.22)[a] | 2 (0.22)[a] | 3 (0.33)[a] | 22 (2.44)[a] |
| | **Winter** | 45 | 0 (0.00) | 0 (0.00) | 0 (0.00) | 0 (0.00) | 0 (0.00) |
| | **Spring** | 1531 | 67 (4.37)[b] | 2 (0.13)[a] | 7 (0.45)[a] | 5 (0.33)[a] | 53 (3.46)[b] |
| | **Summer** | 2678 | 122 (4.55)[b] | 4 (0.15)[a] | 10 (0.37)[a] | 9 (0.34)[a] | 99 (3.69)[b] |
| Total | | **5152** | 218 (4.23) | 08 (0.15)[a] | 19 (0.37)[b] | 17 (0.33)[ab] | 174 (3.38)[c] |
| **Srinagar** | **Autumn** | 409 | 13 (3.17)[a] | 1 (0.24)[a] | 2 (0.49)[a] | 0 (0.00) | 10 (2.44)[b] |
| | **Winter** | 11 | 0 (0.00) | 0 (0.00) | 0 (0.00) | 0 (0.00) | 0 (0.00) |
| | **Spring** | 943 | 34 (3.60)[b] | 1 (0.11)[a] | 4 (0.42)[a] | 4 (0.42)[a] | 25 (2.65)[b] |
| | **Summer** | 1627 | 60 (3.68)[c] | 1 (0.06)[a] | 4 (0.24)[a] | 4 (0.24)[a] | 51 (3.13)[b] |
| Total | | **2990** | 107 (3.57) | 03 (0.10)[a] | 10 (0.33)[b] | 08 (0.27)[ab] | 86 (2.88)[c] |
| **Grand Total** | | **12103** | 488 (4.03) | 16 (0.13)[a] | 42 (0.34)[b] | 35 (0.28)[b] | 395 (3.26)[c] |

Figures in the parenthesis indicate % prevalence

Prevalence values (in parenthesis) of cercaria in a row across columns bearing different small case superscript for a particular district differ significantly

(Fig 8). FC28 showed similarity ranging from 99.8% to 98.7% with *Fasciola gigantica* Kashmir isolate (JF323866), *Fasciola gigantica* Egypt isolate (KF791537), *Fasciola gigantica* Bangalore (HM776945), Kolkata (HM126479), Izzatnagar (JF323865) and Vietnam (MF099787) isolates, *Fasciola gigantica* Burkina Faso (AJ440785) and Santiago (AJ439739) isolates, *Fasciola hepatica* Spain isolate (AJ439738) and *Fasciola gigantica* Karnataka isolate (MN148544) (Fig 8). FZ28 isolate showed similarity ranging from 98.0% to 96.3% with *Fasciola gigantica* Kolkata (HM126479), Bangalore (HM776945) and Vietnam (MF099787) isolates, *Fasciola gigantica* Santiago (AJ439739), Burkina Faso (AJ440785) and *Fasciola hepatica* Spain (AJ439738) isolates, *Fasciola gigantic* Izzatnagar isolate (JF323865), *Fasciola gigantica* Karnataka isolate (MN148544), *Fasciola gigantica* Kashmir isolate (JF323866) and *Fasciola gigantica* Egypt isolate (KF791537) (Fig 8). FA28 showed 99.2% and 97.4% similarity with FC28 and FZ28 isolates, respectively, while as FC28 showed 97.5% similarity with FZ28 isolate (Fig 8). For FA28 isolate, 2 nucleotide polymorphisms were observed at the loci 10 and 603, while as FC28 isolate showed 4 nucleotide polymorphisms at loci 6, 7, 10 and 603. FZ28 showed polymorphisms from loci 566–578 and 594 except at 572.

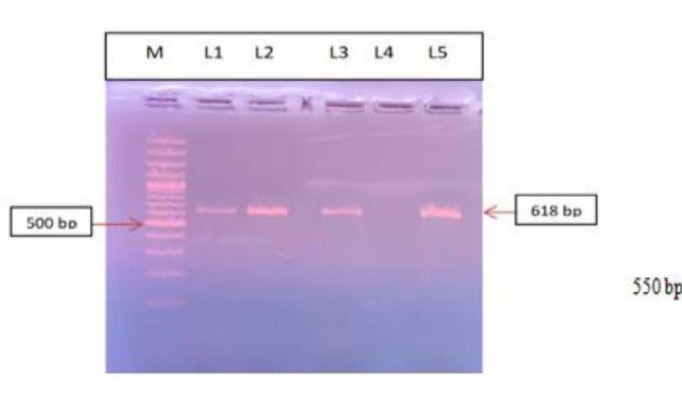

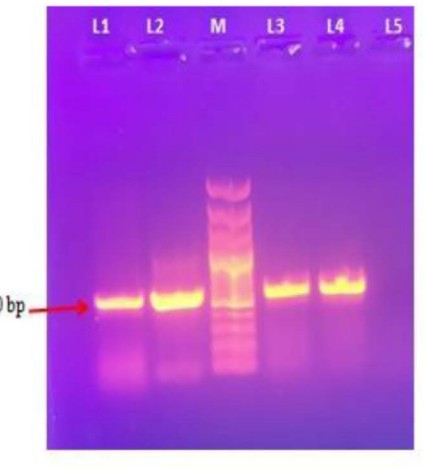

Lane M: 100bp plus DNA marker
Lane 1, 2, 3, 5: 28S rDNA gene
Lane 4: Negative control

28S rDNA gene of gymnocephalus cercaria

Lane M:100bp plus DNA marker
Lane 1, 2, 3, 4: ITS-2 region
Lane 5: Negative control

ITS-2 region of rDNA of gymnocephalus cercaria

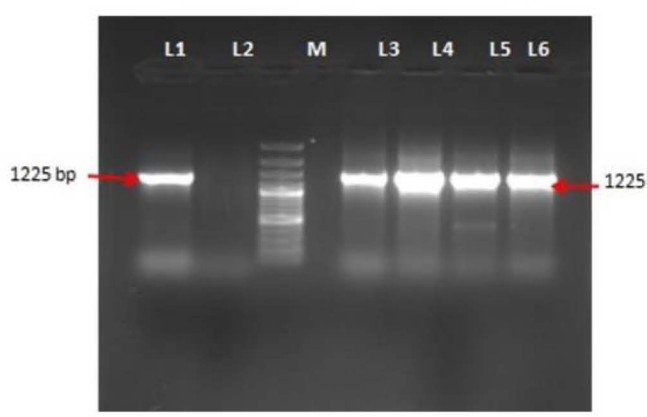

Lane M: 100bp plus DNA marker
Lane 1, 3, 4, 5, 6: 28S rDNA gene
Lane 2: Negative control

28S rDNA gene of furcocercous cercaria

Lane M: 100bp plus DNA marker
Lane 1, 3, 4, 5, 6: ITS-2 region
Lane 2: Negative control

ITS-2 region of rDNA of furcocercous cercaria

**Fig 7. PCR amplification of gymnocephalus and furcocercous cercaria.**

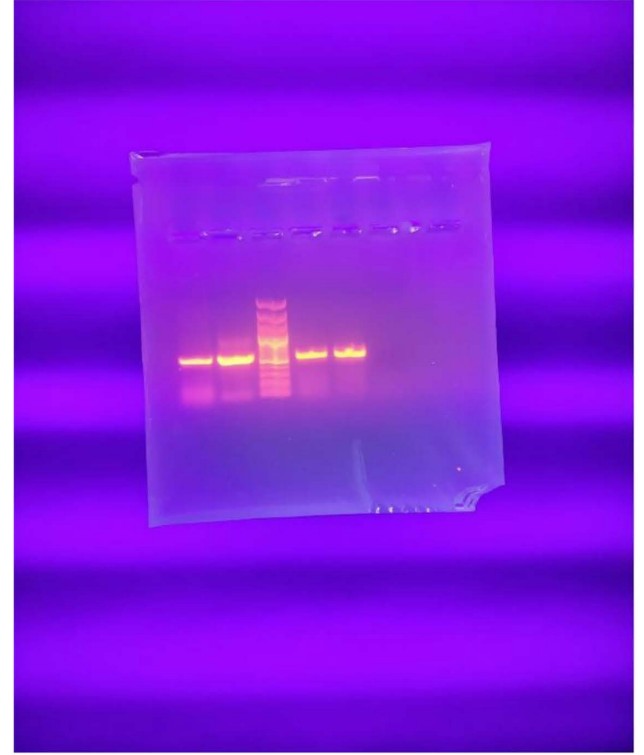

**ITS-2 region of rDNA of gymnocephalus cercaria**

28S rDNA gene of gymnocephalus cercaria

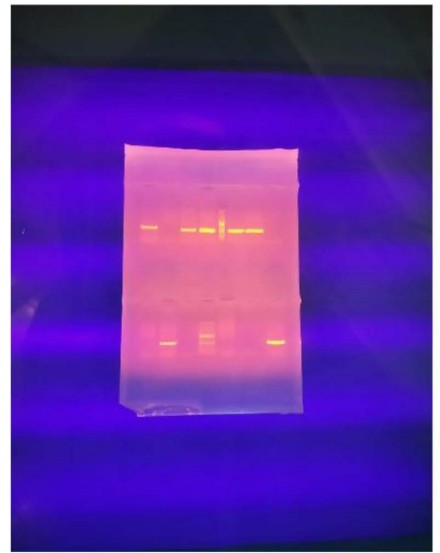
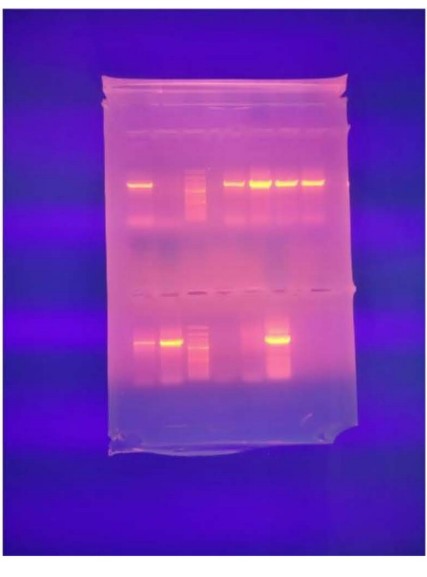
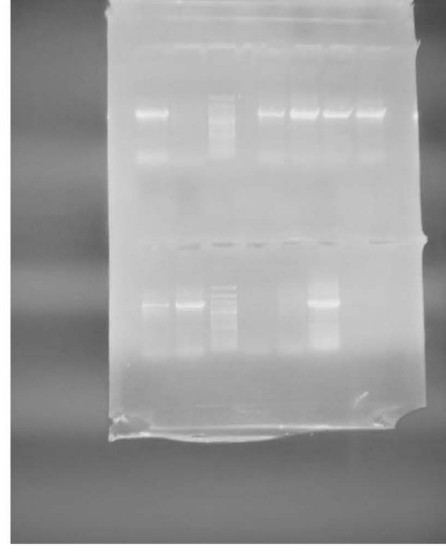

**ITS-2 region of rDNA of furcocercous cercaria**

**28S rDNA gene of furcocercous cercaria (coloured image)**

**28S rDNA gene of furcocercous cercaria (black and white image)**

**Fig 7.** Continued.

| Percent Identity | | | | | | | | | | | | | | | |
|---|---|---|---|---|---|---|---|---|---|---|---|---|---|---|---|
| | 1 | 2 | 3 | 4 | 5 | 6 | 7 | 8 | 9 | 10 | 11 | 12 | 13 | | |
| 1 | | 100.0 | 99.8 | 99.8 | 99.8 | 99.8 | 99.5 | 99.8 | 98.7 | 99.4 | 98.9 | 99.0 | 97.9 | 1 | F.gigantica AJ439739 Santiago Island |
| 2 | 0.0 | | 99.8 | 99.8 | 99.8 | 99.8 | 99.5 | 99.8 | 98.7 | 99.4 | 98.9 | 99.0 | 97.9 | 2 | F.gigantica AJ440785 Burkina Faso Origi |
| 3 | 0.2 | 0.2 | | 100.0 | 100.0 | 100.0 | 99.3 | 100.0 | 98.9 | 99.5 | 99.0 | 99.2 | 98.0 | 3 | F.gigantica HM126479 Kolkata |
| 4 | 0.2 | 0.2 | 0.0 | | 100.0 | 100.0 | 99.3 | 100.0 | 98.9 | 99.5 | 99.0 | 99.2 | 98.0 | 4 | F.gigantica HM776945 Bangalore |
| 5 | 0.2 | 0.2 | 0.0 | 0.0 | | 99.1 | 99.3 | 100.0 | 98.9 | 99.5 | 99.0 | 99.2 | 97.2 | 5 | F.gigantica JF323865 IzzatNagar |
| 6 | 0.2 | 0.2 | 0.0 | 0.0 | 0.0 | | 97.3 | 100.0 | 99.5 | 99.5 | 99.8 | 99.8 | 96.4 | 6 | F.gigantica JF323866 Kashmir |
| 7 | 0.3 | 0.3 | 0.5 | 0.5 | 0.5 | 0.2 | | 99.3 | 98.2 | 98.8 | 99.0 | 99.5 | 96.3 | 7 | F.gigantica KF791537 Egypt |
| 8 | 0.2 | 0.2 | 0.0 | 0.0 | 0.0 | 0.0 | 0.5 | | 98.9 | 99.5 | 99.0 | 99.2 | 98.0 | 8 | F.gigantica MF099787 Vietnam |
| 9 | 1.3 | 1.3 | 1.1 | 1.1 | 1.2 | 0.2 | 1.5 | 1.1 | | 98.4 | 97.9 | 98.1 | 96.6 | 9 | F.gigantica MN148544 Karnataka |
| 10 | 0.7 | 0.7 | 0.5 | 0.5 | 0.5 | 0.5 | 1.0 | 0.5 | 1.6 | | 98.5 | 98.7 | 97.9 | 10 | F.hepatica AJ439738 Spain |
| 11 | 0.3 | 0.3 | 0.2 | 0.2 | 0.2 | 0.2 | 0.5 | 0.2 | 1.3 | 0.7 | | 99.2 | 97.4 | 11 | Fasciola FA28 Kashmir Isolate |
| 12 | 0.8 | 0.8 | 0.7 | 0.7 | 0.7 | 0.2 | 0.5 | 0.7 | 1.8 | 1.1 | 0.6 | | 97.5 | 12 | Fasciola FC28 Kashmir Isolate |
| 13 | 1.2 | 1.2 | 1.0 | 1.0 | 1.0 | 1.1 | 1.4 | 1.0 | 2.2 | 1.2 | 0.8 | 1.3 | | 13 | Fasciola FZ28 Kashmir Isolate |

*Divergence* (left vertical axis label)

**Fig 8. Sequence pair wise distances of 28S rDNA of *Fasciola* isolates (FA28, FC28 and FZ28).**

FA28 differed from FC28 isolate at 6, 7, 9 and 10 nucleotide positions. FA28 and FC28 isolates differed from FZ28 isolate at loci 10, 534, 566–578 except at 572. There was addition of 1 nucleotide at position 10 in case of FA28 and FC28 isolates, and deletion of 8 nucleotides at 566, 567, 568, 574, 575, 576, 577 and 578 loci in case of FZ28 isolate as compared to published sequences of 28S rDNA of *Fasciola* spp. taken in the present study.

The phylogenetic trees were constructed by two approaches *viz.* maximum likelihood approach based on Kimura 2-parameter model and neighbour joining approach separately as well as jointly for all the isolates. The results revealed that the samples belonging to FA28 and FC28 isolates of Kashmir clustered separately into single group with boot strap value 42, while as FZ28 isolate formed a separate group (Fig 9). The gymnocephalus cercariae isolates FA28 and FC28 were thus identified as cercarial stages of *Fasciola gigantica* and FZ28 isolate as *F. hepatica* (Table 8, Fig 9).

**Amplification and nucleotide sequence analysis of ITS-2 region of *Fasciola spp.*** Out of 16 samples, 4 representative samples (FA2, FB2, FC2 and FH2) were subjected to PCR and got amplified with expected product size of 550 bp (Fig 7). Out of these isolates, only 1 representative isolate (FC2) was sent for sequencing. The sequence obtained after sequencing of this amplicon was aligned with the published sequences of ITS-2 region of rDNA gene of *Fasciola* spp. (Accession Nos. JN541193, KX198626, MH048702, KX198616, KX013554, KF667378, KF667377, KF644586, JN541198, JN541197, JN541196, JN541195 and JN541194) (Fig 10).

The total length of the sequenced ITS-2 region of rDNA of *Fasciola* isolate FC2 was 551 bp. The nucleotide sequence of FC2 isolate showed similarity ranging from 99.8% to 91.5% with the *Fasciola hepatica* Kashmir isolate (KX198626), *Fasciola gigantica* Kashmir isolate (KX198616), *Fasciola gigantica* Kashmir isolate (MH048702), *Fasciola gigantica* NE (KF667377 and KF667378) isolates, *Fasciola gigantica* Chennai isolate (KF644586), *Fasciola gigantica* Kashmir isolate (JN541193), *Fasciola gigantica* Kashmir (JN541195 and JN541196) isolates, *Fasciola gigantic* Bangalore isolate (JN541198), *Fasciola gigantica* Kashmir isolate (JN541194), *Fasciola gigantica* Mizoram isolate (KX013554) and *Fasciola gigantic* Bangalore isolate (JN541197) (Fig 10). For *Fasciola* FC2 isolate, no nucleotide polymorphism was seen as compared to published sequences of ITS-2 region of rDNA of *Fasciola* spp. taken in the present study. There were nucleotide substitutions at positions 378 (C→T), 417 (T→C), 423 (T→C), 481 (A→G) and presence of nucleotide T at 475 position which confirmed it be *Fasciola hepatica*.

The phylogenetic trees were constructed by two approaches *viz.* maximum likelihood approach based on Kimura 2-parameter model and neighbour joining approach separately as well as jointly for all the isolates revealed that the FC2

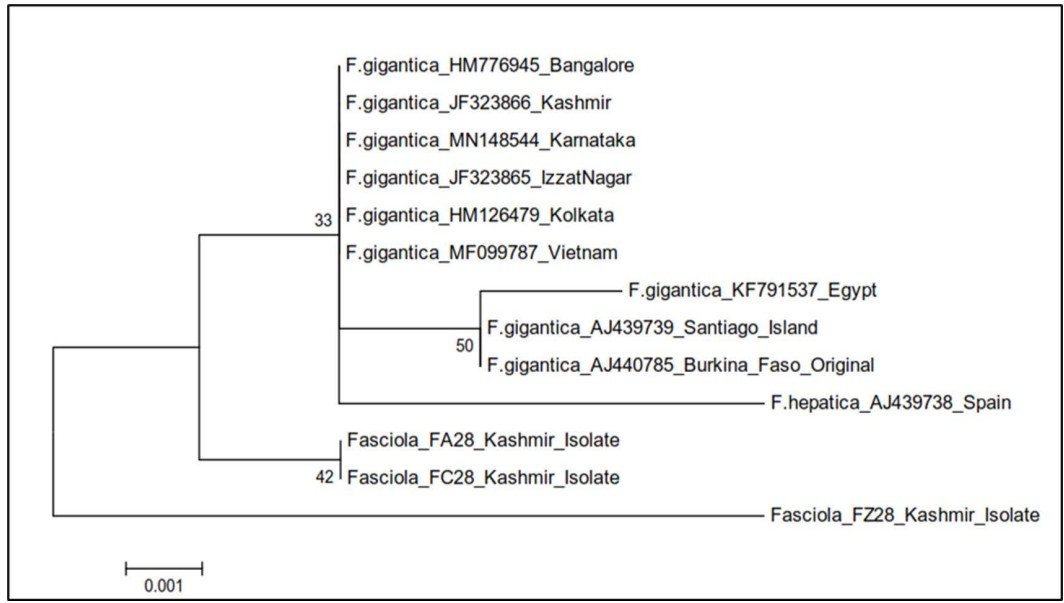

**Fig 9. Phylogenetic tree analysis of *Fasciola* isolates (FA28, FC28 and FZ28) based on 28S rDNA.**

**Table 8. Cercarial isolates of Central Kashmir.**

| S. No | Isolate | Area screened | Snail involved | Cercariae type | rDNA region | Identified cercariae of trematode parasite | Accession No. |
|---|---|---|---|---|---|---|---|
| 1 | FA | Ganderbal | *Lymnaea auricularia* | Gymnocephalus | 28S | *Fasciola hepatica* | PV082153.1 |
| 2 | FC28 | Srinagar | *Lymnaea auricularia* | Gymnocephalus | 28S | *Fasciola gigantica* | PV082152. |
| 3 | FZ 28 | Ganderbal | *Lymnaea auricularia* | Gymnocephalus | 28S | *Fasciola gigantica* | PV082151.1 |
| 4 | FC 2 | Budgam | *Lymnaea auricularia* | Gymnocephalus | ITS 2 | *Fasciola hepatica* | PV082157.1 |
| 5 | B1 | Ganderbal | *Lymnaea stagnalis* | Echinostome | 28S | *Moliniella anceps* | PV082161.1 |
| 6 | BD13 | Budgam | *Lymnaea stagnalis* | Echinostome | 28S | *Echinoparyphium recurvatum* | PV082164.1 |
| 7 | Gy | Ganderbal | *Gyraulus* | Echinostome | 28S | Echinostomatidae spp. | PV082165.1 |
| 8 | B11 | Ganderbal | *Lymnaea brevicauda* | Echinostome | ITS 2 | *Echinoparyphium recurvatum* | PV082166.1 |

isolate and *F. hepatica* Kashmir (KX198626) clustered separately into single group with boot strap value 88 (Fig 11). The gymnocephalus cercaria (FC2) was thus identified as cercarial stage of *Fasciola hepatica* (Table 8, Fig 12).

**Furcocercous cercaria Amplification and identification of furcocercous cercaria.** Out of 35 morphologically identified cercarial samples, 5 representative samples (28S, 28B, 28SC, 28S3 and 28B2) were subjected to PCR for identification of schistosome cercariae by amplification of 28S rDNA gene. The PCR products were of the size of 1225 bp (Fig 7). The purified amplicons of 4 samples (28S, 28B, 28SC and 28S3) were then custom sequenced and the raw sequences thus obtained were analyzed using BLAST search of NCBI for homology. Two cercariae (28B and 28B2) showed highest homology with *Cotylurus marcogliesei* which is a trematode parasite of domestic and wild ducks, pigeons, etc. and other two (28S and 28S3) showed highest homology with *Diplostomum pseudospathaceum*, which is a fish trematode. They belong to the families Strigidae and Diplostomidae respectively and the cercariae of these parasites are also furcocercous in nature. Hence no cercaria of family schistosomatidae was recorded in the present study.

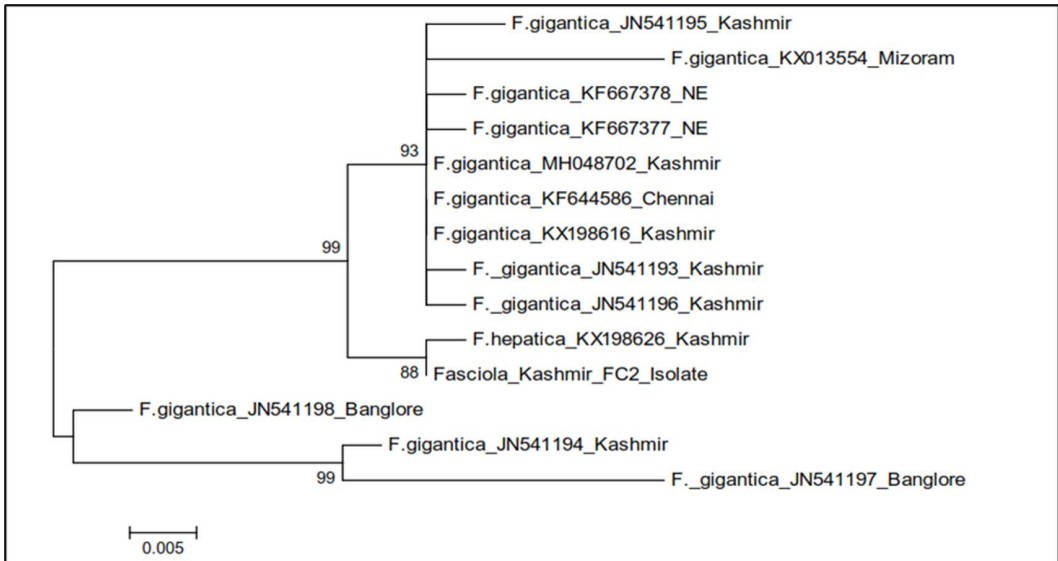

| Percent Identity | | | | | | | | | | | | | | | | |
|---|---|---|---|---|---|---|---|---|---|---|---|---|---|---|---|---|
| | 1 | 2 | 3 | 4 | 5 | 6 | 7 | 8 | 9 | 10 | 11 | 12 | 13 | 14 | | |
| 1 | ■ | 97.2 | 92.2 | 99.7 | 95.0 | 97.5 | 99.4 | 99.4 | 98.6 | 95.0 | 91.4 | 99.2 | 99.2 | 93.1 | 1 | F.gigantica JN541193 Kashmir |
| 2 | 1.4 | ■ | 99.8 | 97.9 | 98.2 | 94.6 | 97.8 | 97.8 | 97.5 | 96.7 | 91.5 | 97.0 | 97.0 | 94.8 | 2 | Fasciola Kashmir FC2 Isolate |
| 3 | 1.8 | 0.2 | ■ | 84.8 | 98.0 | 85.7 | 97.4 | 97.4 | 95.5 | 91.4 | 86.8 | 91.7 | 91.7 | 90.1 | 3 | F.hepatica KX198626 Kashmir |
| 4 | 0.3 | 1.1 | 1.5 | ■ | 86.9 | 94.6 | 99.4 | 99.4 | 100.0 | 95.0 | 91.2 | 99.7 | 99.4 | 93.1 | 4 | F.gigantica MH048702 Kashmir |
| 5 | 0.3 | 0.8 | 1.0 | 0.2 | ■ | 88.1 | 99.6 | 99.6 | 98.3 | 90.1 | 86.2 | 94.7 | 94.5 | 88.1 | 5 | F.gigantica KX198616 Kashmir |
| 6 | 2.5 | 3.8 | 3.1 | 2.9 | 1.9 | ■ | 96.7 | 96.7 | 98.3 | 92.5 | 88.4 | 97.2 | 97.0 | 90.6 | 6 | F.gigantica KX013554 Mizoram |
| 7 | 0.6 | 1.3 | 1.4 | 0.6 | 0.4 | 3.2 | ■ | 99.6 | 99.7 | 94.8 | 90.9 | 99.4 | 99.2 | 92.8 | 7 | F.gigantica KF667378 NE |
| 8 | 0.6 | 1.3 | 1.4 | 0.6 | 0.4 | 3.2 | 0.4 | ■ | 99.7 | 94.8 | 90.9 | 99.4 | 99.2 | 92.8 | 8 | F.gigantica KF667377 NE |
| 9 | 0.3 | 1.1 | 1.5 | 0.0 | 0.0 | 1.7 | 0.3 | 0.3 | ■ | 94.1 | 90.4 | 98.6 | 98.3 | 92.1 | 9 | F.gigantica KF644586 Chennai |
| 10 | 3.1 | 2.8 | 3.3 | 2.8 | 3.0 | 5.5 | 3.1 | 3.1 | 2.9 | ■ | 93.6 | 94.7 | 94.7 | 97.2 | 10 | F.gigantica JN541198 Banglore |
| 11 | 7.0 | 6.7 | 6.7 | 6.7 | 7.0 | 9.5 | 7.0 | 7.0 | 6.9 | 4.6 | ■ | 91.1 | 91.1 | 95.9 | 11 | F.gigantica JN541197 Banglore |
| 12 | 0.6 | 1.4 | 1.8 | 0.3 | 0.3 | 2.8 | 0.6 | 0.6 | 0.3 | 3.1 | 7.0 | ■ | 99.2 | 92.8 | 12 | F.gigantica JN541196 Kashmir |
| 13 | 0.8 | 1.7 | 2.1 | 0.6 | 0.6 | 3.1 | 0.8 | 0.8 | 0.6 | 3.4 | 7.3 | 0.8 | ■ | 92.8 | 13 | F.gigantica JN541195 Kashmir |
| 14 | 4.9 | 4.6 | 4.5 | 4.6 | 4.8 | 7.3 | 4.9 | 4.9 | 4.8 | 2.5 | 2.5 | 4.9 | 5.2 | ■ | 14 | F.gigantica JN541194 Kashmir |
| | 1 | 2 | 3 | 4 | 5 | 6 | 7 | 8 | 9 | 10 | 11 | 12 | 13 | 14 | | |

**Fig 10. Sequence pairwise distances of *Fasciola* isolate (FC2) based on ITS-2 region of rDNA.**

**Fig 11. Phylogenetic tree analysis of *Fasciola* isolate (FC2) based on ITS-2 region of rDNA.**

Further identification was tried by amplifying ITS-2 region of another four representative cercarial samples (ITS2–19, ITS2-D3, ITS2-SC and ITS2-S3) which got amplified with product size of 500 bp (Fig 7). The purified amplicons after sequencing again revealed that two cercariae (ITS2-D3 and ITS2–19) showed highest homology with *Cotylurus marcogliesei* and two (ITS2-SC and ITS2-S3) showed highest homology with *Diplostomum pseudospathaceum*. Thus, validating the results obtained by amplification of 28S rDNA gene.

**Amphistome cercaria. Amplification and identification of amphistome larval stage:** In the present study none of the snails released amphistome cercaria. Further confirmation was done by amplifying larval DNA from snail tissue. About 60 *Indoplanorbis exustus* snails were subjected to PCR for confirming the presence of larval stages of amphistomes in

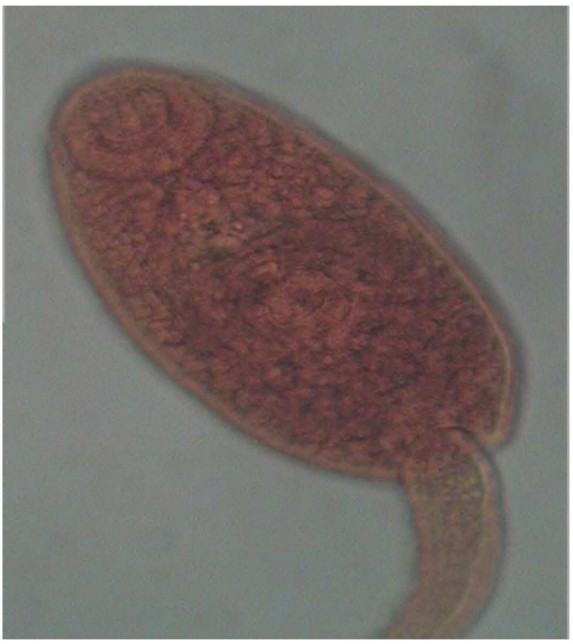

**Cercaria of *Fasciola gigantica***

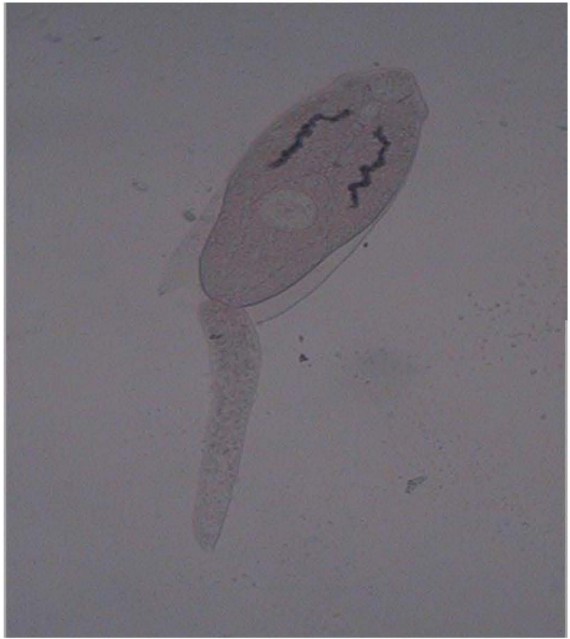

**Cercaria of *Fasciola hepatica***

**Fig 12. Gymnocephalus cercaria identified as *Fasciola gigantica* and *Fasciola hepatica* based on molecular analysis.**

these snails by amplification of 28S rDNA gene. The PCR products obtained were of the size of only 300 bp against the expected size of 1200 bp (S1 Raw Image). The purified amplicons of only 4 representative samples (A8, B8, C8 and D8) were then custom sequenced and these sequences were analyzed using BLAST search of NCBI for homology. It was observed that the larval stages present in these snails belonged to *Paratamatam iquitosiensis*, which is a blood fluke of turtle and *Plagiorchis elegans* which is a trematode parasite of mammals particularly mice and does not belong to Paramphistomatidae family. Hence no cercariae/other larval stages of paramphistomes were recorded in the study.

Similarly, the ITS-2 region of rDNA of the samples (A2, B2, C2 and D2) was amplified using gene specific primers which got amplified with product size of 500 bp (S1 Raw Image). The purified amplicons were sent for sequencing but could not be further processed because of non-retrieval of the amplicons by the company.

**Echinostome cercaria. Amplification and nucleotide sequence analysis of 28S rDNA gene of *Echinostoma* spp.:** Out of 42 morphologically identified cercarial samples, 4 representative samples (B1, BD13, BD22 and GY) were subjected to PCR which got amplified with product size of 554 bp (S1 Raw Image). The sequences obtained after sequencing of 3 of these amplicons (B1, BD13 and GY) were aligned with the published sequences of 28S rDNA gene of *Echinostoma* spp. (Accession Nos. KY513157, KP065591, MK482501, AY222246, DQ471888, EF470905, EF470908, KF894680, MK321656, MK321667, MK321668 and KT956921) (Fig 13).

The total length of the sequenced 28S rDNA gene of *Echinostoma* isolate B1 was 618 bp, BD13 was 617 bp and that of GY isolate was 616 bp. The nucleotide sequence of B1 isolate showed similarity ranging from 95.8% to 91.7% with *Moliniella anceps* USA isolate (KT956921), Echinostomatidae USA isolate (MK321668), Echinostomatidae USA isolate (MK321656) and *Echinoparyphium recurvatum* Czech Republic isolate (KY513157), Echinostomatidae USA isolate (MK321667), *Echinostoma revolutum* Thailand isolate (KF894680), *Echinostoma caproni* USA isolate (MK482501), *Echinostoma bolschewense* Czech Republic isolate (KP065591) and *Echinostoma revolutum* USA isolate (DQ471888), *Echinostoma revolutum* USA isolates (EF470905 and EF470908) and *Echinostoma revolutum* London isolate (AY222246) (Fig 13). BD13 isolate showed similarity ranging from 98.4% to 94.1% with *Echinoparyphium recurvatum* Czech Republic isolate (KY513157) and Echinostomatidae USA isolate (MK321656), Echinostomatidae USA isolate (MK321668), Echinostomatidae USA isolate (MK321667), *Echinostoma bolschewense* Czech Republic isolate (KP065591) and *Echinostoma*

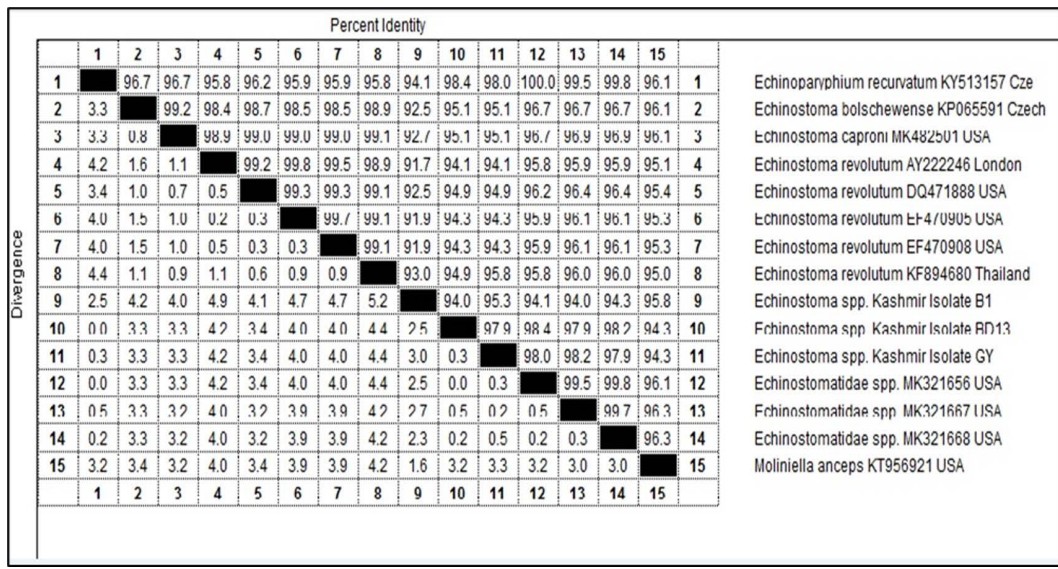

**Fig 13. Sequence pair wise distances of *Echinostoma* isolates (B1, BD13 and GY) based on 28S rDNA.**

*caproni* USA isolate (MK482501), *Echinostoma revolutum* USA isolate (DQ471888) and *Echinostoma revolutum* Thailand isolate (KF894680), *Echinostoma revolutum* USA isolates (EF470905 and EF470908) and *Moliniella anceps* USA isolate (KT956921) and *Echinostoma revolutum* London isolate (AY222246) (Fig 13). GY isolate showed similarity ranging from 98.2% to 94.1% with Echinostomatidae USA isolate (MK321667), *Echinoparyphium recurvatum* Czech Republic isolate (KY513157) and Echinostomatidae USA isolate (MK321656), Echinostomatidae USA isolate (MK321668), *Echinostoma revolutum* Thailand isolate (KF894680), *Echinostoma bolschewense* Czech Republic isolate (KP065591) and *Echinostoma caproni* USA isolate (MK482501), *Echinostoma revolutum* USA isolate (DQ471888), *Echinostoma revolutum* USA isolates (EF470905 and EF470908) and *Moliniella anceps* USA isolate (KT956921) and *Echinostoma revolutum* London isolate (AY222246) (Fig 13). B1 isolate showed 94.0% and 95.3% similarity with BD13 and GY isolates of Kashmir respectively, while as BD13 isolate showed 97.9% similarity with GY isolate (Fig 13). For B1 isolate, 6 nucleotide polymorphisms were observed at the loci 9, 219, 223, 225, 368 and 404, while as 2 nucleotide polymorphisms were observed in BD13 isolate at loci 9 and 553. GY isolate showed polymorphisms at two loci, i.e., 9 and 609. B1 isolate differed from BD13 isolate at 121, 136, 196, 219, 222, 223, 225, 228, 282, 333, 355, 362, 368, 372, 375, 404, 412, 555 and 611 nucleotide positions. B1 isolate differed from GY isolate at 121, 136, 196, 219, 222, 223, 225, 228, 282, 333, 355, 362, 368, 372, 375, 404, 412, 415, 517, 555 and 611 nucleotide positions. BD13 isolate differed from GY isolate at loci 414, 515, 553 and 609 nucleotide positions as compared to published sequences of 28S rDNA of *Echinostoma* spp. taken in the present study.

The phylogenetic trees were constructed by two approaches *viz.* maximum likelihood approach based on Hasegawa-Kishina-Yano parameter model and neighbour joining approach separately as well as jointly for all the isolates. The results revealed that the samples belonging to isolate BD13, *Echinostoma* spp. USA isolate (MK321656) and *Echinoparyphium recurvatum* Czech Republic isolate clustered separately into single group with boot strap value 63, isolate GY clustered with *Echinostoma* spp. USA isolate (MK321667) separately into single group with boot strap value 57, while as isolate BI and *Moliniella anceps* isolate of USA clustered separately into single group with boot strap value 87 (Fig 14). The cercariae were thus identified as cercarial stages of species of family Echinostomatidae (GY), *Echinoparyphium recurvatum* (BD13) and *Moliniella anceps* (B1) (Table 8, Fig 14).

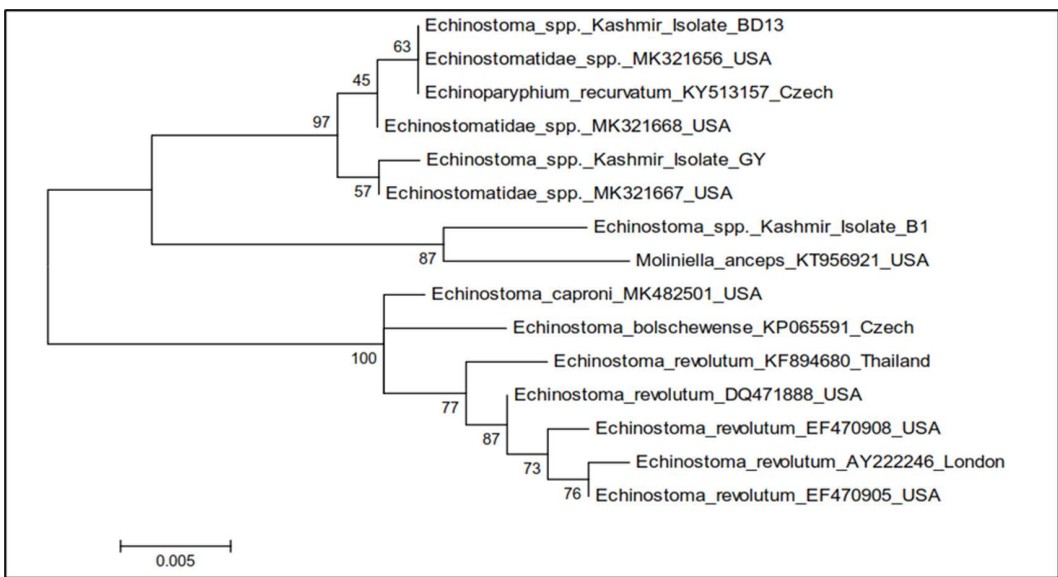

**Fig 14. Phylogenetic tree analysis of *Echinostoma* isolates (B1, BD13 and GY) based on 28S rDNA.**

**Amplification and nucleotide sequence analysis of ITS-2 region of *Echinostoma spp.*** Out of 42 morphologically identified cercarial samples, 6 representative samples (B1, BD13, BD22, B33, B44 and GY) were subjected to PCR, which got amplified with product size of 588 bp (S1 Raw Image). Out of these, only one isolate (B1) was sent for sequencing for further confirmation. The sequences obtained after sequencing of this amplicon was aligned with the published sequences of ITS-2 region of rDNA gene of *Echinostoma* spp. Accession Nos. (KJ542640, KJ435271, AF067850, AF026791 and GQ463126) (Fig 15).

The total length of the sequenced ITS-2 region rDNA of *Echinostoma* spp. isolate B1 was 632 bp which showed similarity ranging from 96.0% to 86.7% with *Echinoparyphium recurvatum* Mexico isolate (KJ435271), *Echinostoma* spp. Australia isolate (AF026791), *Echinoparyphium mordvilkowi* Liphuania isolate (KJ542640), *Echinostoma trivolis* USA isolate (GQ463126) and *Echinostoma revolutum* USA isolate (AF067850) (Fig 15). For *Echinostoma* spp. Kashmir isolate B11, 7 nucleotide polymorphisms were seen at loci 18, 117, 118, 218, 360, 437 and 494. There was addition of one nucleotide at position 18 as compared to published sequences of ITS-2 region of rDNA of *Echinostoma* spp. taken in the present study (Fig 15). The phylogenetic trees were constructed by maximum likelihood approach based on Kimura 2- parameter model and neighbour joining approach separately as well as jointly for all the isolates revealed that the isolate B11, *Echinostoma* spp. Australia isolate and *Echinoparyphium recurvatum* Mexico isolate clustered separately into single group with boot strap value 54 (Fig 16). The cercaria was thus identified as cercarial stages of *Echinoparyphium recurvatum* (Table 8, Fig 17).

| | | Percent Identity | | | | | | |
|---|---|---|---|---|---|---|---|---|
| | **1** | **2** | **3** | **4** | **5** | **6** | | |
| **1** | ■ | 92.5 | 86.5 | 92.0 | 91.4 | 86.9 | **1** | Echinoparyphium mordvilkowi KJ542640 Li |
| **2** | 4.0 | ■ | 86.3 | 97.8 | 96.0 | 87.5 | **2** | Echinoparyphium recurvatum KJ435271 Mex |
| **3** | 7.7 | 8.6 | ■ | 87.0 | 86.7 | 97.9 | **3** | Echinostoma revolutum AF067850 USA |
| **4** | 3.6 | 0.4 | 7.6 | ■ | 93.8 | 87.6 | **4** | Echinostoma spp. AF026791 Australia |
| **5** | 4.5 | 3.1 | 7.6 | 3.4 | ■ | 87.1 | **5** | Echinostoma spp. Kashmir Isolate B1 |
| **6** | 7.4 | 7.5 | 1.1 | 6.8 | 7.3 | ■ | **6** | Echinostoma trivolvis GQ463126 USA |
| | **1** | **2** | **3** | **4** | **5** | **6** | | |

(Left axis label: Divergence)

**Fig 15. Sequence pair wise distances of ITS-2 region of rDNA of *Echinostoma* isolate (B1).**

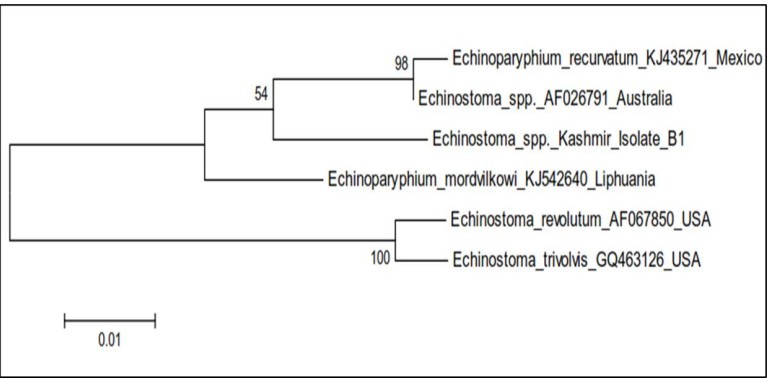

**Fig 16. Phylogenetic tree analysis of *Echinostoma* isolate (B1) based on ITS-2 region of rDNA.**

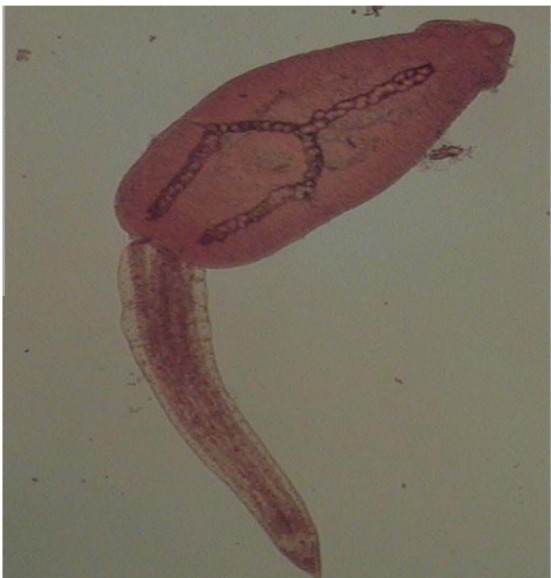

**Cercaria of *Echinoparphyium recurvatum***

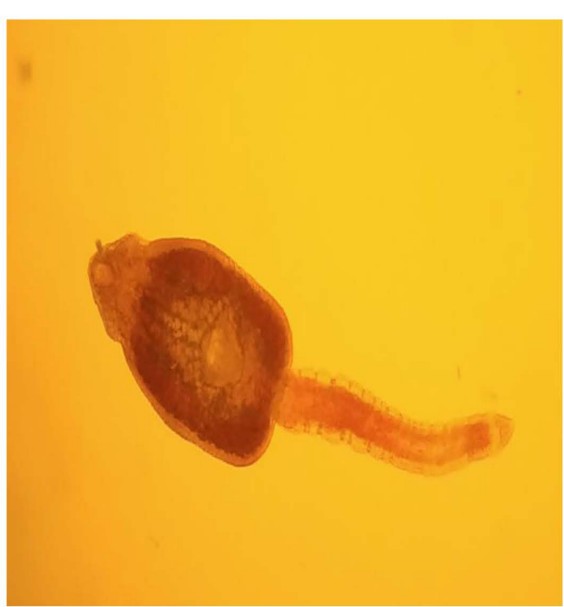

**Cercaria of *Molinella anceps***

**Fig 17. Echinostome cercaria identified as *Echinoparphyium recurvatum* and *Molinella anceps* based on molecular analysis.**

## Discussion

### Overall prevalence of cercariae in snails

In the present study, overall prevalence of cercariae was observed to be 4.03%. The results of the present investigation are analogous to the findings of Devkota et al. [35] who recorded 4.3% infection of snails with trematode cercariae at

Chitwan District of Nepal; Islam et al. [36] who observed 5.3% snails shedded cercariae in Mymensingh, Bangladesh; Chotananarth et al. [13] who reported an overall prevalence of larval trematodes to be 4.7% in freshwater snails of Nakhon Province, Thailand; Dunghungzin and Chontananarthn [37] who observed the prevalence of cercariae to be 2.45%.

Gymnocephalus cercaria, echinostome cercaria, furcocercous cercaria and xiphido-cercaria were recorded in the present study. Cercarial stages like furcocercous and echinostomes were also reported by Sharif et al. [38] in lymnaeid snails of Central areas of Mazandaran, Iran; Namchote et al. [39] recorded parapleurophocercous cercariae, xiphido-cercariae, furcocercous cercariae, echinostome cercariae and cotylomicrocercous cercariae from snails of Central and East coast of the Gulf of Thailand. Ramitha and Vasandakumar [40] recorded occurrence of furcocercous, echinostome, xiphidocercous cercariae in freshwater snails of Malabar and Chotananarth et al. [13] reported echinostome, furcocercous and xiphido-cercariae in freshwater snails of Nakhon Province, Thailand. The variation in the results might be due to differences in the number of snail samples examined and presence of susceptible snails in the study areas. The reason for not recording amphistome cercaria could be either collection of snails after the release of cercaria or collection of snails from areas where contact between animal faeces and snails are rare as well as collection of snails before they have picked up the infection. *Indoplanorbis exustus* is considered to be the most suitable intermediate host for amphistomes and total number of *Indoplanorbis exustus* snails collected from 3 districts was 644, which means that about 215 snails were collected from each district. The lower number of snails collected and lower level of infection in animals as reported [41–43] may probably explain non recording of amphistome cercaria in the present study.

Among cercariae, highest prevalence was recorded for xiphido-cercaria (3.26%) followed by echinostome cercaria (0.34%), furcocercous cercaria (0.28%) and lowest for gymnocephalus cercaria (0.13%). In each district, prevalence of these cercaria was almost similar to the observed overall prevalence of cercariae in snails of Central Kashmir. These results approximate to the findings of Hussein and Khalifa [44] who stated that the infection with xiphido-cercaria was the most prevalent in Qena Governorate, Upper Egypt; Devkota et al. [35] who recorded 0.96% xiphido-cercariae infection in snails from Chitwan District, Nepal; Mbaya [45] who recorded 0.55% *Fasciola* spp. cercaria in snails of Borno state, Nigeria; Fernandez and Hamann [46] who recorded 0.40% infection of snails with furcocercous cercaria in northeastern Argentina; Dunghungzin et al. [47] who reported the highest prevalence of xiphido-cercaria (0.81%) followed by echinostome cercaria (0.27%) and furcocercous cercaria (0.09%) in freshwater snails of Phra Nakhon Province, Thailand.

At District level, highest prevalence of cercariae was recorded in District Ganderbal (4.23%) followed by District Budgam (4.11%) and District Srinagar (3.57%), the difference being statistically non-significant (p > 0.05) between districts. Chigwena et al. [48] observed the occurrence of larval trematodes to be 6.6% in freshwater snails of Highveld and lowveld areas of Zimbabwe; Nithiuthai et al. [49] recorded 3.04% snails infected with cercariae in the main water reservoir of Huai Thalaeng District, Thailand; Tigga et al. [50] recorded overall prevalence of cercariae as 7.33% in freshwater snails of Ranchi District, Jharkhand and Anucherngchai et al. [51] who reported an overall infection rate of cercaria to be 5.90% in snails of Chao-Phraya basin, Thailand. The variation in prevalence of cercariae in snails with regard to the findings of other workers may be due to the variation in the number of snails examined, level of trematode infection in the animals, duration of the study period, meteorological factors that govern the breeding, life span of infected snails and development of different developmental stages within the snails as well as access of livestock and domestic birds to water bodies and also for using of raw cow dung as manure etc. The reason for low prevalence rate of snail borne trematodal infection can be attributed to the fact that the maximum animals are either stall fed or grazed in apple orchards, hence minimizing the chances of exposure of freshwater snails to the trematode eggs present in the faeces. A low prevalence of *Fasciola* spp. in sheep from various districts of Kashmir Valley has been reported by [41,52]. This reduction in incidence in animals is probably because of extensive prophylactic measures which are being adopted in letter and spirit against this extremely important trematodal disease of sheep in Jammu and Kashmir. So, reduction in the prevalence rates of this trematode will automatically reduce infection rate in snails.

## Prevalence of cercariae in different snails

*L. stagnalis* and *L. lagotis* snails released echinostome, furcocercous and xiphido-cercaria. *L. brevicauda* snails were positive for echinostome cercaria and xiphido-cercaria. *L. auricularia* snails, released gymnocephalus cercaria and xiphido-cercaria. *B. troscheli*, *L. luteola* and *P. acuta* snails were positive for xiphido-cercaria only. *G. ladacensis* snails were positive for echinostome cercaria and furcocercous cercariae. *I. exustus*, *G. pankogensis* and unidentified snail did not release any cercaria. One possible explanation for this observation might be the resistance of these snail species to trematode infection, a factor which was also noted earlier [53]. Similar types of cercariae from respective snails were reported by workers *viz.*, Laman et al. [54] observed avian schistosomes in *G. parvus* snails of Indiana; Sharma et al. [55] stated that *L. auricularia* supported the development of *F. gigantica* under laboratory conditions in Kashmir Valley; Loy and Haas [56] who reported echinostome, schistosome and xiphido-cercariae from *L. stagnalis* snails in a pond system of Southern Germany; Morely et al. [57] reported xiphido-cercariae from *Bithynia tentaculata* snails collected from Lower Thames Valley, London; Singh et al. [58] recorded *Fasciola* spp. from *Lymnaea auricularia* in Faizabad, U.P; Garg et al. [59] recorded larval stages of *F. gigantica* (1.57%) from *Lymnaea auricularia* in plains of north India; Sharif et al. [33] reported furcocercous and echinostome cercariae from lymnaeid snails of Central areas of Mazandaran, Iran; Soldanova et al. [60] recorded echinostome, schistosome and xiphidocercariae from *L. stagnalis* and *L. auricularia* snails of Ruhr river, Germany; Imani-Baran et al. [61] reported xiphido-cercaria, furcocercous and echinostome cercariae from *Lymnaea* snails particularly *L. gedrosiana* snails of Iran. The results of the present study agree with Anderson and May [62], who attributed the low prevalence of cercarial shedding among natural populations to the direct consequence of parasite-induced host mortality. The other reason could also be attributed to early collection of snails before they have picked up infection or collection of snails from the locations which may not be easily accessible to animals. Application of flukicides to the animals may cause death of flukes before they attain maturity resulting in either release of fewer eggs or embryonic mortality of eggs. Regular heaping of faeces outside animal sheds may also cause destruction of fluke eggs by heat of fermentation before application of such faeces to the fields.

## Overall seasonal prevalence of cercariae in snails

Highest prevalence of cercariae was recorded in summer (4.28%) followed by spring (4.05%) and autumn (3.22%) season, the difference being statistically significant ($p < 0.05$) between autumn and summer; spring and summer seasons. Hiekal and El-Sokkary [63] observed that seasonal incidence of cercariae was higher in summer followed by spring and autumn season. Karimi et al. [64] observed peak incidence of infection in summer followed by autumn and winter and spring. Nkwengulila and Kigadye [65] reported the prevalence of digeneans higher during the dry season and lower in the rainy season in snails of Ruvu Basin, Tanzania. Imani-Baran et al. [61] reported higher cercarial infection of snails in summer season in Iran. Higher prevalence in summer season might be because of favourable climatic conditions for development and breeding of snails, as a result of which they develop and multiply fast, increasing their populations quickly, which increases the chance of contact between fluke egg/ miracidia and snails. The availability of good macrophyte vegetations in summer season enhance the growth and development of snails and acquire large size, which help in maturation of larval stages quickly. Large sized snails are also able to tolerate the pathological effect of trematode larvae, thus increasing their chances of survival which might be another factor of more cercarial shedding during spring and summer seasons. No cercaria was recorded during winter season which might be due to lower temperature that inhibits the development of the larval stages of trematodal parasites within snails.

Gymnocephalus cercaria showed higher prevalence in autumn followed by spring and summer season. Highest prevalence of echinostome and furcocercous cercaria was recorded in spring followed by summer and autumn season, while as xiphido-cercaria showed higher prevalence in summer followed by spring and autumn season. None of the cercariae was recovered in winter season. Chowdhury et al. [66] also reported prevalence of gymnocephalus cercaria higher in

July-August followed by September-October and May-June. Islam et al. [67] reported developmental stages of *Fasciola gigantica* higher in rainy season followed by summer and lowest during winter season. Niaz et al. [68] observed furco-cercous cercariae highest in autumn (17.6%) followed by summer (17.0%), winter (10.0%) and lowest in spring season (4.27%). The variation in the observed prevalence could be due to the differences in the number of snails examined, the timing of collection of infected snails and managemental practices adopted for rearing of livestock and disposal of faeces. During winter (December to February), development and hatching of eggs ceases or takes long time due to low temperature. Therefore, snails are not infested during this period and hence no cercaria infection was recorded in snails until spring. On the onset of spring season, owing to increase in temperature and filling of ponds, rice fields, and other water bodies with rain and snowmelt water and also availability of vegetation allows snail eggs to hatch and develop faster as food source is increased. Also, the animals are allowed to move out of houses, they feed and drink near the places which are breeding sites of these snails. The faeces come in contact with snail infested water and chance of snail getting infected with miracidia of flukes increases. The development of cercarial stages of flukes within the snails takes some time. These favorable climatic conditions persist from Mid-April onwards for the development and hatching of fluke eggs and other developmental stages of flukes within the snails. Thus, cercariae infestation in the snails begins to increase from April and May continue until October as some snails may pick up infection during late summer and release cercariae until October. The occurrence of echinostome and furcocercous cercariae is likely due to the presence of migratory and residential bird species and to several varieties of fishes and reptiles in and around the water bodies. Many bird species act as final host for strigeoid and clinostomatoid digeneans. Hechinger and Lafferty [69] demonstrated consistent, positive and significant associations between final host bird communities and trematode communities in intermediate host snail populations. The reason for echinostome and furcocercous cercariae infection higher in spring and summer may be due to favorable climatic conditions in the area that make it attractive for the wild birds to migrate and settle during mid spring to late summer. On the other hand, there is a possibility that snails infected by bird's larval trematodes can live throughout the winter by hibernation, there by releasing cercariae during spring season. This phenomenon is supported by the results of studies on the behaviour of *L. stagnalis* in response to temperature [70]. This low prevalence of infection could also be due to low parasite pressure, simply making contact between miracidia and snails a rare occurrence. Our findings of lower trematode cercarial diversity in this case may be influenced by factors such as study seasons and the distribution and abundance of definitive hosts as has also been observed [71]. In all the 3 districts the observed seasonal prevalence of cercariae was almost similar to the overall seasonal prevalence.

### Molecular study

**Gymnocephalus cercaria.** In order to confirm that this cercarial type acts as developmental stage of *Fasciola* spp., the sequences of 28S and ITS-2 region of rDNA of gymnocephalus cercariae were amplified using gene specific primers [29,30] respectively. In this study, the cercarial isolates FA28, FC28 and FZ28 showed amplification of 28S rDNA gene of *Fasciola* spp. with 97.9 to 99.8% nucleotide sequence homology with different isolates of *Fasciola* spp. registered in the Gene Bank. The total length of the sequenced 28S rDNA gene of *Fasciola* isolates FA28 and FC28 was 619 bp, while for isolate FZ28, it was only 610 bp.

Sequence alignment of the 28S rDNA from available *Fasciola* isolates of India (Izzatnagar, Bangalore and Kolkata), Egypt, Vietnam, Santiago, Burkina Faso and Spain showed a distinct variation (A/G) at 284th nucleotide position group-ing these *Fasciola* into two distinct clades of 284A and 284G lineages of 28S rDNA. *Fasciola* isolates FA28, FC28 and FZ28 in the present study grouped in 284th A lineage of 28S rDNA, which in our study is 269th nucleotide position as only the partial sequence of 28S rDNA gene, corresponding to 16–663 bp (618 bp) region, was amplified using forward and reverse primers. The results are supported by the findings of Raina et al. [30] who also reported that *Fasciola* isolates of India fall under 284 A lineage of *F. gigantica* for 28S rDNA. Bauri et al. [20] also reported that sequence analysis of 28S rDNA showed a distinct variation (A/G) at 284th nucleotide and phylogenetic analysis clustered snail isolate of *F. gigantica*

into 284A clade along with other isolates of the parasite reported from different parts of India. Teofanova et al. [72] proposed existence of 2 lineages for 28S rDNA in *F. hepatica* in Eastern Europe based on polymorphism at 105th nucleotide position. Many investigators have found no differences in the 28S rDNA gene within a species or not at a significant level to provide evidence of genetic diversity [29,73]. Only few interspecific nucleotide variations have been found between *F. hepatica* and *F. gigantic* [29]. The reason for this polymorphism may be a common genetic pool of all Indian isolates of *Fasciola* spp. and geographical barrier between India and other European and African countries as well as no transmigration of animals across borders.

The phylogenetic tree based on 28S in the present study showed that FA28 and FC28 isolates of Kashmir clustered separately into single group with boot strap value 42, while as FZ28 Kashmir isolate formed a separate group, suggesting that two separate species of *Fasciola* are able to infect animals in Central Kashmir. The gymnocephalus cercarial isolates FA28 and FC28 were accordingly identified as cercarial stages of *Fasciola gigantic* and FZ28 isolate as *F. hepatica.* For FA28 isolate, 2 nucleotide polymorphisms were observed, while as FC28 isolate showed 4 nucleotide polymorphisms. FZ28 showed polymorphisms from loci 566–578 and 594 except at 572. FA28 differed from FC28 isolate at 4 nucleotide positions. FA28 and FC28 isolates differed from FZ28 isolate at 14 positions. Walker et al. [74] also reported that there are 4 positions at which nucleotide variation occurred among African, Indian and European *Fasciola* flukes which included 105, 130, 283 and 547. These polymorphisms recorded in the present study need detailed investigations.

In the present study, the cercarial isolate (FC2) showed amplification of ITS-2 region of *Fasciola* with nucleotide sequence exhibiting 91.5% to 99.8% similarity with different isolates of *Fasciola* registered in the Gene Bank. The total length of the sequenced ITS-2 region of rDNA of *Fasciola* isolate FC2 was 551 bp. On construction of phylogenetic tree, the results revealed that the FC2 isolate and *F. hepatica* Kashmir isolate (KX198626) clustered separately into single group with boot strap value 88. The gymnocephalus cercaria (FC2) was accordingly identified as cercarial stage of *Fasciola hepatica*. The nucleotide substitutions at positions 378 (C→T), 417 (T→C), 423 (T→C), 481 (A→G) and presence of nucleotide T at 475 positions confirmed it as *Fasciola hepatica*. This is in line with the studies carried out by 72 Adlard et al. [75]; Agatsuma et al. [76]; Ali et al. [77] (2008) and Raina et al. [30] who also found that *F. hepatica* differed from *F. gigantica* in ITS-2 at six positions. For FC2 isolate, no nucleotide polymorphism was seen as compared to published sequences of ITS-2 region of rDNA of *Fasciola* taken in the present study. Semyenova et al. [78] and Erensoy et al. [79] have also not recorded any nucleotide change among *F. hepatica* collected from different locations such as Russia, Belrus, Ukraine, Armenia and Turkmenistan and Turkish *F. hepatica* samples, respectively. Our findings are in line with Ghavami et al. [80], who reported that ITS-2 region of fasciolids is highly conserved with no intraspecific variation and does not vary in length or nucleotide composition when compared with different isolates. Khalifa et al. [81] also reported no nucleotide variation in ITS-2 sequences of *Fasciola* when compared with ITS-2 sequences from Egypt, Turkey, Japan, Vietnam, Tunisia, Spain and Iran. The explanation seems to be that, repeated DNA sequences, such as rDNA have been subjected to concerted evolution, which tends to homogenize sequences among individuals and among populations [82]. The absence of intraspecific variation within the species is indicative of highly conserved nature of rDNA internal transcribed spacers as has been demonstrated earlier [83]. The phylogenetic tree showed a close relationship of FC2 isolate with other isolates from Kashmir [84]. Farjallah et al. [85] also found a close relationship of Tunisian and Algerian isolates of *Fasciola hepatica* with those from Niger, Turkey, Egypt, Ireland and Iran. In the present study, no intermediate *Fasciola* species cercaria was recorded and this is in line with the findings of Shakeebah [84], who did not report intermediate forms of *Fasciola* spp. in Central Kashmir.

**Furcocercous cercaria.** In the present study, none of the cercarial isolates was identified as developmental stage of *Schistosoma* spp. after sequencing of the amplified products. The amplification of 28S rDNA and ITS-2 region of furcocercous cercaria using gene specific primers meant for amplification of these genes in *Schistosoma* spp. [31,32] could be due to non-specific binding of primers to the DNA of closely related genera of *Schsitosoma* spp. (*Cotylurus*

*marcogliesei* and *Diplostomum pseudospathaceum*) in the absence of *Schistosoma* genomic DNA. These two identified cercarial stages belong to the families Strigidae and Diplostomidae respectively which are also furcocercous in nature.

**Amphistome cercaria.** In the present study none of the snails released amphistome cercaria. Further confirmation was done by amplifying larval DNA from snail tissue. After sequencing the amplified products, it was observed that the larval stages present in these snails do not belong to Paramphistomatidae family. The amplification of 28S rDNA and ITS-2 region of suspected snail tissue using gene specific primers meant for amplification of these genes [33,86] could be due to non-specific binding of primers to the DNA of closely related genera of amphistomes in the absence of amphistome genomic DNA.

**Echinostome cercaria.** The 28S rDNA gene and ITS-2 region of rDNA of Echinostome cercariae were amplified using gene specific primers [31,86]. In this study, the cercarial isolates B1, BD13 and GY showed amplification of 28S rDNA gene of Echinostomes with 91.7 to 98.4% nucleotide sequence homology with different isolates of Echinostomes registered in the Gene Bank. The total length of the sequenced 28S rDNA gene of Echinostomatidae isolate B1 was 618 bp, BD13 was 617 bp and that of GY isolate was 616 bp. The Echinostome cercarial isolates B1, BD13 and GY were accordingly identified as cercarial stages of *Moliniella anceps*, *Echinoparyphium recurvatum* and Echinostomatidae spp., respectively. Phylogenetic tree construction revealed that isolate BD13, *Echinostoma* spp. USA isolate (MK321656) and *Echinoparyphium recurvatum* Czech Republic isolate clustered separately into single group with boot strap value 63, isolate GY clustered with *Echinostoma* spp. USA isolate (MK321667) separately into single group with boot strap value 57, while as isolate BI and *Moliniella anceps* isolate of USA clustered separately into single group with boot strap value 87. Laidemitt et al. [87] on phylogenetic analyses found 17 clades of Echinostomes in East Africa with clade 1 representing *Echinostoma caproni*, clade 4–6 Echinostomatidae spp. and clade 7 *Echinoparyphium* spp. In the present study a dendrogram was constructed for 28S sequence data with 1000 replicates for the purpose of molecular identification. The dendrogram acquired from these results can be separated into 3 clades including Echinostomatidae species/*Echinoparyphium recurvatum*, *Moliniella anceps* and *Echinostoma* spp. (*E. caproni*, *E. bolschewense* and *E. revolutum*). For B1 isolate, 6 nucleotide polymorphisms were observed, while as 2 nucleotide polymorphisms were observed in BD13 and GY isolates. The sequences of the 3 distinct isolates (B1, BD13 and GY) of Echinostome cercaria differed at various nucleotide positions. There are 19 nucleotide differences between B1 and BD13; 21 between B1 and GY and 4 nucleotide differences between BD13 and GY. In the present study, B1 isolate showed prominent differences at 121 (C→T), 136 (G→A), 196 (A→C), 355 (C→T), 362 (T→C), 372 (T→C), 375 (T→C) and 612 (insertion of C) with respect to other Echinostomatidae spp. and is same as reported earlier in *Moliniella anceps* (Accession No. KT956921). The results suggested that isolate B1 belongs to the *Moliniella anceps*, isolate GY to Echinostomatidae spp. and isolate BD13 to genus *Echinoparyphium* and there is high possibility of it being a sibling species or a species within *E. recurvatum* complex as it clustered and closely aligned with *E. recurvatum* in the phylogenetic tree.

In the present study, the cercarial isolate (B11) showed amplification of ITS-2 region of *Echinostoma* with nucleotide sequence exhibiting 86.7% to 96.0% similarity with different isolates of Echinostomes registered in the Gene Bank. The total length of the sequenced ITS-2 region of rDNA of isolate B11 was 632 bp. On construction of phylogenetic tree, the results revealed that the isolate B1, *Echinostoma* spp. Australia isolate and *Echinoparyphium recurvatum* Mexico isolate clustered separately into single group with boot strap value 54. The cercaria was thus identified as cercarial stages of *Echinoparyphium recurvatum*. Chontananarth et al. [13] constituted dendrogram of trematode cercariae based on ITS-2 sequence was separated into 6 clades among which two included Echinostomatida/Echinostmatidae and Ehinostomatida/ Philophthalmidae. Tkach et al. [88] at higher taxonomic levels, by phylogenetic analysis provided strong support for eight family level lineages in Echinostomatoidea. In their study clade 1 comprised 17 genera sampled within the Echinostomatidae, Psilostomidae, Cathaemascidae and Rhopaliidae including *Echinostoma*, the type genus of the Echinostomatidae. It splits into two major sub-clades, one including *Echinostoma*, *Molineilla*, *Echinoparyphium*, *Neoacanthoparyphium*, *Patagifer*, *Artyfechinostomum* and *Hypoderaeum*. For isolate B11, 7 nucleotide polymorphisms were observed. Saijuntha

et al. [89] also characterized two morphologically unidentified Echinostomes with one of them showing 99% identity to and clustered with genus *Echinoparyphium*, whereas the other was located in the revolutum group. The authors also concluded that 28S rDNA and ITS-2 regions are suitable molecular markers for genetic characterization and phylogenetic analysis of Echinostomes. In *E. mordvilkowi* prominent nucleotide changes occur at 291(A→G) and 462 (G→A) and in *E. recurvatum* at 566 (T→C) with respect to other Echinostomes. In our study for isolate B11 all changes were encountered, i.e., at 291, 462 and 566, thus indicating that B11 isolate belongs to the genus *Echinoparyphium* with high possibility of it being a sibling species or a species with *E. recurvatum* as it showed 96.0% similarity with it as against only 91.4% with *E. mordvilkowi*.

## Conclusion

The proportion of snails that release cercariae and the number of cercariae released from each snail play important roles in the transmission of trematodes from the snail host. Knowledge of the snails and cercariae released by them could be used to formulate control strategy to reduce the burden of trematode parasites in animals. The study made it very clear that molecular characterization employing ITS-2 and 28S rDNA sequences are reliable approach for genetic differentiation of cercarial stages of trematodes. The phylogenetic taxonomy of echinostomes is still unclear and molecular diversity found in this study shall be helpful in understanding their morphological, biological and molecular diversity for clarifying their taxonomic position. Presence of avian Echinostome cercaria in the snails indicates that the infection is prevalent in the domesticated birds but exact incidence needs to be worked out.

## Supporting information

**S1 Raw Image. PCR amplification of amphistome and echinostome cercaria.**
(DOCX)

## Acknowledgments

The authors are highly thankful to the Dean, FVSc. and AH, SKUAST-K, Shuhama in arranging funds for smooth conductance of research work. The authors are also thankful to Divisional staff for assisting in the laboratory and sincere thanks are due to Mr. Gh. Rasool for his help throughout the research programme. The help rendered by him cannot be forgotten and is worth to remember.

## Author contributions

**Conceptualization:** Rafiq A. Shahardar.

**Data curation:** Kamal H. Bulbul.

**Formal analysis:** Idrees M. Allaie.

**Investigation:** Rafiq A. Shahardar.

**Methodology:** Kamal H. Bulbul.

**Resources:** Aiman Ashraf.

**Software:** Shahana Tramboo.

**Validation:** Showkat A. Shah.

**Visualization:** Asif H. Khan.

**Writing – original draft:** Zahoor Ahmad Wani.

**Writing – review & editing:** Shabir A. Rather.

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
