## [Decision Letter · Decision Letter 0]

Dear Dr. Wani,

Thank you for submitting your manuscript to PLOS ONE. After careful consideration, we feel that it has merit but does not fully meet PLOS ONE’s publication criteria as it currently stands. Therefore, we invite you to submit a revised version of the manuscript that addresses the points raised during the review process.

**ACADEMIC EDITOR:**

The authors need to include the morphological identification of snails and good figures for cercariae

We look forward to receiving your revised manuscript.

Kind regards,

Shawky M Aboelhadid, PhD

Academic Editor

PLOS ONE

Additional Editor Comments (if provided):

Reviewers' comments:

Reviewer's Responses to Questions

**Comments to the Author**

1. Is the manuscript technically sound, and do the data support the conclusions?

Reviewer #1: No

2. Has the statistical analysis been performed appropriately and rigorously?

Reviewer #1: I Don't Know

3. Have the authors made all data underlying the findings in their manuscript fully available?

Reviewer #1: No

4. Is the manuscript presented in an intelligible fashion and written in standard English?

Reviewer #1: Yes

Reviewer #1: The manuscript titled "Seasonal Dynamics and Molecular Phylogenetic Studies on Cercariae in Central Zone of Kashmir Valley" presents a detailed investigation into the prevalence, identification, and ecological dynamics of cercariae in snails from the Kashmir region.

The manuscript examines 12,103 snails across multiple districts and seasons, offering robust insights into cercariae prevalence.

Selection of studied snails is not clear and morphological identification of snails is missing. All snail species could not be vectors for trematodes. And it is emphasised not to use only DNA sequences but also morphology in combination with genetic data, however, in the present study the morphology is characterised only very superficially and the classification of cercarial morphotypes is not clear.

In the results, authors not explain morphology of each cercarial types. Although molecular studies were performed, but the morphology should be considered.

Furthermore, it is necessary to record voucher samples in a museum collection for further examination of snails, just like you have to archive genetic data these days.

In molecular analysis, only you used 28S rDNA and ITS-2 targeted genes, with these genes, closely related species cannot be separated, only a mitochondrial gene can do that.

However, the results have much more information that was discussed in the manuscript and more effort to discuss exactly what putative species were found, particularly with the genetic data or even the snail host. Below are some suggestions to help tease out more of the data from what was a major collection and screening of snails. Thus, make the most of the hard work to get the data. Also please voucher samples in a museum collection, just like you have to archive genetic data these days.

Moreover, the sequences and identifications provided are not linked to morphological data. A table with the newly obtained sequence accession numbers, with identification, host, morphotypes, and locality is missing; there are only data in the trees, however, it is impossible to see from which particular host.

In phylogenetic analysis, bootstrap values are very low (statistically, only 70% or above as a good support).

In Introduction,

The introduction needs to be more on point of what was the reason for undertaking this study with seasonal dynamic (although you highlight in the title).

In Materials and Methods,

For study areas, you should include more details about geographical information of locations (e.g GPS information).

Morphological examination of snails should be included although you include some of snail species in results.

DNA extraction - it is not clear how the cercariae were sampled - were the cercariae taken from one snail specimen? or were they pooled from more snails of the same species? For DNA, morphologically identical cercariae should be taken only from one snail specimen.

Did the authors perform an autopsy of snails or repeat shedding experiment? Only one shedding examination usually does not allow all infected snails to be recorded. For the snails that did not release cercaria the time that you examined, how did you sure that the snails were not be infected. You may need to provide more information about the examination of the cercaria in the materials and methods such as how often did you do/repeat shedding the cercaria and how long that the snails were kept for this experiment?

Using 28S rDNA and ITS is good to distinguish genera and some species apart, but closely related species cannot be separated, only a mitochondrial gene can do that. Plus cox1 is used a lot for barcoding and while it has drawbacks, it is still a good proxy. ITS is also a very conservative gene and not good for species IDs. Thus, if this was the easiest combination of genes to use, then make that clear and why you did not use mitochondrial genes.

However, in the present study, I do not see the link between the morphologically characterised specimens and the sequences, i.e. there is no table provided showing a list of sequence numbers together with identification of the isolates, their host(s) and locality data; there are only data in the tree (Fig.), however, it is impossible to see from which particular host.

One of the most important questions about your data, did you keep vouchered specimens for the snail samples? The only way for studies such as these to be repeatable is the availability of vouchered material available for re-examination. Furthermore, the identification of invertebrate hosts, including gastropods is notoriously difficult and constantly changing on top of the fact that population studies have shown lots of cryptic species.

Boakes et al 2010. Distorted views of biodiversity: spatial and temporal bias in species occurrence data. PLOS Biology 8: e1000385

Carlson et al. 2020. What would it take to describe the global diversity of parasites? Proceedings of the Royal Society B 287: 20201841

Harmon et al. 2019. Parasites lost: using natural history collections to track disease change across deep time. Front. Ecol. Environ. 17: 157-166.

Hoberg et al. 2009. Why museums matter: a tale of pinworms (Oxyuroidea: Heteroxynematidae) among pikas in the American West. Journal of Parasitology 95: 490-501.

Shultz et al. 2021. Natural history collections are critical resources for contemporary and future studies of urban evolution. Evolutionary Applications 14: 233-247

And this one in particular highlights the need for host vouchers:

Thompson et al. 2021. Preserve a voucher specimen! The critical need for integrating natural history collections in infectious disease studies. mbio 12: e02698-20

In Figures and graphs,

Visual representations of data (e.g., graphs showing seasonal prevalence) should be used.

Pictures resolution should be upgraded.

In phylogenetic analysis (Fig 4, 9 and 11), sequences from this study should be highlighted. More data should be used for phylogenetic analysis as well as outgroups should be included. Moreover, bootstrap values are very low (statistically, only 70% or above as a good support). Please see below reference:

Zander RH. 2004. Minimal values of reliability of Bootstrap and Jackknife proportions, Decay index, and Bayesian posterior probability. PhyloInformatics 2:1-13.

Hillis DM and Bull JJ. 1993. An Empirical Test of Bootstrapping as a Method for Assessing Confidence in Phylogenetic Analysis. Systematic Biology, 42 (2):182–192.

https://academic.oup.com/sysbio/article-abstract/42/2/182/1730933?login=false

**Do you want your identity to be public for this peer review?** For information about this choice, including consent withdrawal, please see our Privacy Policy

Reviewer #1: No

---

## [Author Response · Author response to Decision Letter 1]

27 Apr 2025

Response to Academic Editor

Comment 1:

The authors need to include the morphological identification of snails and good figures for cercariae.

Response 1: Thank you very much for your support and guidance.

Identification of snails with cercaria images: Identification of snails has been mentioned under collection and identification of snail’s sub-section mentioned at line no. 102 of revised manuscript which is highlighted in yellow and good cercariae images that were best among the photographs taken are already placed at respective places.

Journal Requirements

Comment 1.

Response 1: The paper has been prepared strictly as per PLOS guidelines.

Comment 2.

In your Methods section, please provide additional information regarding the permits you obtained for the work. Please ensure you have included the full name of the authority that approved the field site access and, if no permits were required, a brief statement explaining why.

Response 2: Since these snails were collected from naturally occurring water bodies, therefore no work permit was required.

Comment 3: Please update your Data Availability statement in the submission form accordingly.

Response 3: Data availability statement provided in the revised manuscript at line no. 834 highlighted in yellow.

Comment 4.

PLOS requires an ORCID iD for the corresponding author in Editorial Manager on papers submitted after December 6th, 2016. Please ensure that you have an ORCID iD and that it is validated in Editorial Manager.

Response 4. ORDID ID of corresponding author: https://orcid.org/0000-0002-1352-0334. We have also provided ORCID ID in title page of revised manuscript highlighted in yellow.

Response to Reviewer comments

Comment 1:

Selection of studied snails is not clear and morphological identification of snails is missing. All snail species could not be vectors for trematodes. And it is emphasised not to use only DNA sequences but also morphology in combination with genetic data, however, in the present study the morphology is characterised only very superficially and the classification of cercarial morphotypes is not clear. In the results, authors not explain morphology of each cercarial types. Although molecular studies were performed, but the morphology should be considered.

Furthermore, it is necessary to record voucher samples in a museum collection for further examination of snails, just like you have to archive genetic data these days.

Response 1:

Morphological identification of snails was done from Zoological Survey of India, Kolkata under voucher specimen number (ZSI, Moll: I.R.No.107) and Department of Parasitology, College of Veterinary Sciences, Assam Agricultural University, Khanapara.

However molecular identification is under progress.

Comment 2:

In molecular analysis, only you used 28S rDNA and ITS-2 targeted genes, with these genes, closely related species cannot be separated, only a mitochondrial gene can do that.

Response 2:

Since primers of 28S rDNA and ITS-2 were already available in our laboratory, they were employed to carry out this study. Besides, limited financial support was available, therefore, primers to amplify genes like mitochondrial region were not procured. Further studies which support the fact that 28SrDNA and ITS-2 can be used for taxonomical studies have been considered in this regard and the links of such references are as follows:

https://parasitesandvectors.biomedcentral.com/articles/10.1186/s13071-020-04124-z

https://academic.oup.com/aesa/article/95/2/250/51008

Comment 3:

Moreover, the sequences and identifications provided are not linked to morphological data. A table with the newly obtained sequence accession numbers, with identification, host, morphotypes, and locality is missing; there are only data in the trees, however, it is impossible to see from which particular host.

Response 3:

Morphological identification of cercaria is given at line no. 205 of the revised manuscript.

These morphologically identified cercariae (which are in figure 1 of manuscript) like Gymnocephalus have been identified as Fasciola gigantica and Fasciola hepatica while Echinostome have been identified as Echinoparphyium recurvatum and Molinella anceps (Figures 17 and 18 in revised manuscript) based on molecular analysis, thereby providing a link between morphological study and molecular study.

Thanks for your guidance in this regard. After careful assessment, we have incorporated a table which is highlighted in yellow, containing details as asked in this comment. Please refer to line No. 725 (Table 8) of the revised manuscript.

Comment 4: In phylogenetic analysis, bootstrap values are very low (statistically, only 70% or above as a good support).

Response 4:

Thanks for your valuable comment. The reason for such low bootstrap values is lack of sufficient data. If the dataset used to build the tree is too small, it might not have enough information to reliably infer evolutionary relationships, resulting in low bootstrap values.

Comment 5:

In Introduction,

The introduction needs to be more on point of what was the reason for undertaking this study with seasonal dynamic (although you highlight in the title).

Response 5:

Thanks for your suggestion. The response to this comment is provided in the introduction section of revised manuscript starting from line no. 53 highlighted in yellow.

Seasonal dynamic studies: The previous studies carried out in this part of Kashmir indicate that helminth parasites show a definite seasonal trend and significant differences in prevalences have been recorded. Temperature and vegetation play a definite role in the growth and maturation of snails as a result their population would follow a definite pattern. Increase and decrease in snail population will directly affect the cercarial output from them. In addition to it, snails undergo hibernation when there is an increase in temperature above 300C and decrease in temperature below 100C.

Comment 6:

In Materials and methods,

For study areas, you should include more details about geographical information of locations (e.g GPS information).

Response 6:

Geographical information of location of study areas (GPS information): GPS information of study areas is provided in a tabular form at line no. 96 of revised manuscript.

Comment 7:

Morphological examination of snails should be included although you include some of snail species in results.

Response 7:

Reply to this comment is the same as response of comment 1 above in reviewer comment section of this response sheet.

Comment 8:

DNA extraction - it is not clear how the cercariae were sampled - were the cercariae taken from one snail specimen? or were they pooled from more snails of the same species? For DNA, morphologically identical cercariae should be taken only from one snail specimen.

Did the authors perform an autopsy of snails or repeat shedding experiment? Only one shedding examination usually does not allow all infected snails to be recorded. For the snails that did not release cercaria the time that you examined, how did you sure that the snails were not be infected. You may need to provide more information about the examination of the cercaria in the materials and methods such as how often did you do/repeat shedding the cercaria and how long that the snails were kept for this experiment?

Response 8:

Sample has been taken from a single snail (mentioned in the manuscript at line no.136, highlighted in yellow).

Autopsy or repeated shedding: Snails which did not shed cercariae by shedding method, were subjected to crushing method to observe the presence of any larval stage in them (mentioned at line no.113 in the manuscript).

Comment 9:

Using 28S rDNA and ITS is good to distinguish genera and some species apart, but closely related species cannot be separated, only a mitochondrial gene can do that. Plus cox1 is used a lot for barcoding and while it has drawbacks, it is still a good proxy. ITS is also a very conservative gene and not good for species IDs. Thus, if this was the easiest combination of genes to use, then make that clear and why you did not use mitochondrial genes.

Response 9:

Same as response 2 of reviewer comment 2.

Comment 10:

However, in the present study, I do not see the link between the morphologically characterised specimens and the sequences, i.e. there is no table provided showing a list of sequence numbers together with identification of the isolates, their host(s) and locality data; there are only data in the tree (Fig.), however, it is impossible to see from which particular host.

Response 10:

Same as response 3 of reviewer comment 3.

Comment 11:

One of the most important questions about your data, did you keep vouchered specimens for the snail samples? The only way for studies such as these to be repeatable is the availability of vouchered material available for re-examination. Furthermore, the identification of invertebrate hosts, including gastropods, is notoriously difficult and constantly changing on top of the fact that population studies have shown lots of cryptic species.

Response 11:

Yes. Morphological identification of snails was done from Zoological Survey of India, Kolkata under voucher specimen number (ZSI, Moll: I.R.No.107) and Department of Parasitology, College of Veterinary Sciences, Assam Agricultural University, Khanapara.

However molecular identification is under progress.

Comment 12:

In Figures and graphs,

Visual representations of data (e.g., graphs showing seasonal prevalence) should be used.

Pictures resolution should be upgraded.

Response 12:

Bar diagrams depicting different prevalences has been included in the manuscript (Fig 2,3,4,5 and 6) and highlighted in yellow. Further we have tried our best to upgrade the resolution of pictures and photographs were captured by high resolution microscopes available to us in our department.

Comment 13:

In phylogenetic analysis (Fig 4, 9 and 11), sequences from this study should be highlighted. More data should be used for phylogenetic analysis as well as outgroups should be included.

Response 13:

Sequences from this study are highlighted in yellow as per the comment (i.e. in previous figures with numbers as 4, 9, 11) which are now revised figures with Serial numbers as 9, 14 and 16 respectively, in the phylogenetic trees of revised manuscript.

Your suggestions regarding the usage of more data for phylogenetic analysis as well as outgroups are highly commendable. But unfortunately, data regarding such analyses is not available to us and one of the co-authors, (sequence analyzer) has kept this data with him and is out of touch for quite a long time and also not responding to our mail communication. So, we are sorry about this.

Comment 14:

Moreover, bootstrap values are very low (statistically, only 70% or above as a good support). Please see the reference below

Zander RH. 2004. Minimal values of reliability of Bootstrap and Jackknife proportions, Decay index, and Bayesian posterior probability. PhyloInformatics 2:1-13.

Hillis DM and Bull JJ. 1993. An Empirical Test of Bootstrapping as a Method for Assessing Confidence in Phylogenetic Analysis. Systematic Biology, 42 (2):182–192.

https://academic.oup.com/sysbio/article-abstract/42/2/182/1730933?login=false

Response 14:

Thanks for this guidance.

Since limited funds were available in the Department, only few samples were submitted for outsource sequencing, that is why boot strap value may be 40 in our case.

---

## [Decision Letter · Decision Letter 1]

Seasonal Dynamics and Molecular Phylogenetic Studies on Cercariae in Central Zone of Kashmir valley

PONE-D-24-55166R1

Dear Dr. Wani,

We’re pleased to inform you that your manuscript has been judged scientifically suitable for publication and will be formally accepted for publication once it meets all outstanding technical requirements.

Kind regards,

Shawky M Aboelhadid, PhD

Academic Editor

PLOS ONE

Additional Editor Comments (optional):

Reviewers' comments:

Reviewer's Responses to Questions

**Comments to the Author**

Reviewer #1: All comments have been addressed

2. Is the manuscript technically sound, and do the data support the conclusions?

Reviewer #1: Yes

3. Has the statistical analysis been performed appropriately and rigorously?

Reviewer #1: Yes

4. Have the authors made all data underlying the findings in their manuscript fully available?

Reviewer #1: Yes

5. Is the manuscript presented in an intelligible fashion and written in standard English?

Reviewer #1: Yes

Reviewer #1: After minor revision, the manuscript can be accepted for publication.

Fig. 9 should be moved to supplementary data.

**Do you want your identity to be public for this peer review?** For information about this choice, including consent withdrawal, please see our Privacy Policy

Reviewer #1: No

---

## [Editor Report · Acceptance letter]

PONE-D-24-55166R1

PLOS ONE

Dear Dr. Wani,

I'm pleased to inform you that your manuscript has been deemed suitable for publication in PLOS ONE. Congratulations! Your manuscript is now being handed over to our production team.

Kind regards,

on behalf of

Professor Shawky M Aboelhadid

Academic Editor

PLOS ONE